# Learning the ESG Geometry with Domain Aware Language Models

Kunal Pradeep Pimparkhede [1]   Chirayu Chaurasia [1]   Jatin Roy [2]   Mahesh Mohan M R [1]

## Abstract

Responsible investing aims to generate positive impact across Environment (E), Society (S), and Governance (G), and rating companies along these dimensions is now widespread, making ESG scores highly popular. Allocating retail capital with sustainability in mind could be transformational, yet it remains unclear how individual investors can do so in practice. Current ESG solutions cannot model high-dimensional, multi-modal time series capturing the joint evolution of ESG risks, financial returns, news, and sentiment, even though this domain requires jointly reasoning over distinct numerical signals where both numerical proximity and semantic type must be preserved. To bridge this gap, we introduce a novel domain-aware **representation learning framework** that learns geometry-preserving representations for heterogeneous time series using value-aware tokens with block-wise **orthogonal embeddings**. To capture trajectory-level structure, we introduce **FACET** tokens and train the model using a geometry-preserving loss. The resulting model jointly learns to forecast future values and to organize entities in a representation space that reflects their temporal evolution. Trained on ESG, returns, news, and sentiment, the domain-aware LLM learns a representation space that enables accurate ESG forecasting, trajectory-based grouping, and latent-space search for superior asset selection and downstream application like portfolio rebalancing.

## 1. Introduction

ESG ratings help investors to invest in companies that focus on *Environment* (e.g., actively reduce carbon emissions), *Society* (e.g., promote diversity and fair labor practices), and *Governance* (e.g., uphold transparent management and ethical business conduct). (Yan et al., 2025) highlights that the firms with better ESG ratings also experience a considerable reduction in employee fraud, discrimination, etc. (Lim, 2024) positions ESG ratings and scores as central tools for measuring, forecasting, and managing ESG-related risks and opportunities in finance. It also highlights the absence of a structured framework that integrates ESG dimensions and AI methodologies. (Edhrabooh et al., 2024) identifies gaps such as usage of outdated ML models with ESG Ratings. We present a state-of-the-art solution using large language models (LLMs) for the ESG domain. The proposed domain-aware LLM supports ESG forecasting, trajectory-based company grouping, and asset replacement for portfolio rebalancing. In this work, a trajectory refers to the time-ordered sequence of past and future values of a single variable (e.g., ESG risk, financial return, or sentiment) for a given company. Our design enables the model to reason over multi-modal time series spanning ESG measures, financial returns, textual news and sentiment achieved through domain-aware tokenization and parameter-efficient adaptation of the LLM. Specifically, we introduce domain-aware and trajectory-aware embeddings that allow tokens to be compared within and across heterogeneous time series. Since each company has multiple trajectories, we further introduce special **FACET tokens** whose hidden states represent the geometry of each trajectory. Beyond predictive accuracy, we also evaluate the quality of downstream socio-economic decisions, including portfolio construction and constrained capital allocation on real ESG data. Contributions of this work can be summarized as:

**1. Domain-aware LLMs:** We introduce Value-Aware Domain Tokens and block-wise embeddings that preserve semantic type and numerical geometry while modeling ESG and financial trajectories.

**2. Forecasting and simulation:** We propose an LLM-based forecaster for ESG and individual E/S/G risks that predicts future sustainability trajectories from historical financial and textual signals.

**3. Performance-preserving peer discovery:** We introduce a trajectory-level representation learning method using FACET tokens and show how the learned embedding space enables Superior Asset Population (SAP) construction for downstream applications such as portfolio optimization.

---

[1]Indian Institute of Technology, Kharagpur [2]Vellore Institute of Technology, Chennai. Correspondence to: Kunal Pradeep Pimparkhede <kunal.pimparkhede.24@kgpian.iitkgp.ac.in>.

*Proceedings of the $43^{rd}$ International Conference on Machine Learning*, Seoul, South Korea. PMLR 306, 2026. Copyright 2026 by the author(s).

## 2. Related work

**Importance of ESG:** (Bock et al., 2025) highlights firms with good investor relations tend to receive better ESG ratings. (Fang & Yang, 2025) shows that firms having higher ESG scores are tend to be more innovative and long term development focused. (Zhan et al., 2025; Li & Zhang, 2024) conclude ESG scores significantly elevate green innovation. (Touti et al., 2025) gives bibliometric analysis of around 390 publications stating that ESG ratings are considered as one of the major risk factors in asset pricing mechanism.(Taskin et al., 2025) establishes the feasibility of predicting future ESG Scores from past ESG scores which our work advances.

**ESG Risk Rating and its Forecasting:** (Yang, 2025) concludes that ESG scores' trends and their temporal characteristic are missed by traditional models. (Binzaiman et al., 2024) highlights the importance of machine learning (ML) based approaches for ESG score prediction. Analysis of (Kim et al., 2024) identifies significant interaction between sentiment direction, ESG ratings, and financial performance. (Edhrabooh et al., 2024) identifies gaps in current prediction of ESG scores viz. usage of outdated ML Models, which highlights scope for improvement. Recently there has been many works for time series forecasting using advanced ML models. (Ngee et al., 2024) evaluates multiple BERT models for automated ESG scoring. A major drawback of the existing approaches for ESG forecasting is the lack of numerical awareness of diverse ESG signals and their interrelationship (e.g., risk, returns and news sentiments). Though not specific to ESG, (Miller et al., 2024) conducts a survey of deep learning and foundation models for time series forecasting. In this context, we refer to some recent literature relevant to our method. (Radford et al., 2019) improves accuracy of time series prediction combining capabilities of LSTM with ARIMA. (Ansari et al., 2024) reformulates time series forecasting as a language modeling task by scaling and quantizing real-valued series into discrete tokens and fine-tuning LLM model to enable zero shot probabilistic forecasting. Despite the introduction of numerous LLM based time series models, (Spathis & Kawsar, 2024) shows that LLMs poorly tokenize numerical data. To this end, this paper proposes ESG domain aware numerical embedding to capture structure and interrelationship across different ESG attributes for ESG time series forecasting. This in turn supports various downstream tasks.

**ESG based Portfolio Rebalancing:** (Iglesias-Casal et al., 2025) emphasizes the need for adaptive and crisis aware portfolio rebalancing model that can influence investment decisions. We introduce the first ever ESG based portfolio rebalancing influenced by LLM embeddings. Relevant works related to our approach are as follows: (Keraghel et al., 2024; Petukhova et al., 2025) shows that LLM embeddings represent natural categorical divisions of data in an unsupervised way. (Viswanathan et al., 2024) uses LLM em-

beddings for a variety of text clustering tasks. On ESG based portfolio rebalancing, (Li et al., 2025) introduces investor aware portfolio rebalancing with deep learning to generate profit focused portfolios. (Martínez-Barbero et al., 2024) uses past average of ESG scores to build high ESG rated portfolio, and recommends usage of genetic algorithms. (Vikash et al., 2025) focuses on portfolio diversification to maximize returns using data sourced from Yahoo Finance. Our work forecasts future sustainability rather than relying on past averages and demonstrates integration with profit focused multi-objective portfolio optimizers.

## 3. Methodology

Figure 1 provides the block diagram of our ESG Domain-Aware System, establishing the foundation for a subsequent discussion. **(1)** The inputs to the system broadly are: i) ESG risk time series (TS) including individual Environment, Society, and Governance risks TS, ii) Financial returns TS, iii) News text and sentiments TS. Current portfolio of the investor also serves as a reference point input. The input corpus, covering datasets (i), (ii), and (iii), is sourced from real-world financial data providers, including Yahoo Finance, Financial Modeling Prep (FMP), and LSEG Refinitiv. **(2)** The Kernel component first constructs **Value-Aware Domain Tokens (VADTs)**, where each VADT packages *numerical value* along with *semantic context*; hence our system can identify type of value represented by VADT (e.g., ESG ratings, financial return, news sentiment). Since different trajectories exhibit distinct temporal patterns, we introduce **FACET tokens**, one per trajectory, whose hidden states encode each trajectory's geometry. Part **(3)** shows the data structure produced by the kernel, which also includes an instruction (e.g., predict ESG, ENV, SOC, or GOV) specifying the target series to forecast. **(4)** VADTs have domain-sensitive embeddings, with different token types mapped to mutually orthogonal subspaces. **(5)** To model how VADTs co-occur across companies, we fine-tune the LLM with LoRA using a dual-objective loss that jointly enforces forecasting accuracy and geometric alignment. This enables the model to capture token trajectories within and across companies. The embeddings of trajectory-specific FACET tokens are learned using a DTW-based loss to represent each trajectory's geometry. We store these FACET embeddings in a vector database to support efficient similarity-based retrieval conditioned on specific FACET dimensions. Part **(6)** mentions the outcomes generated by our model to elevate sustainability levels. We provide a representation learning framework that embeds individual company trajectories while preserving numerical value and semantic type. The learned embedding space supports querying to construct **Superior Asset Populations (SAP)** and yields improved convergence in downstream tasks such as portfolio optimiza-

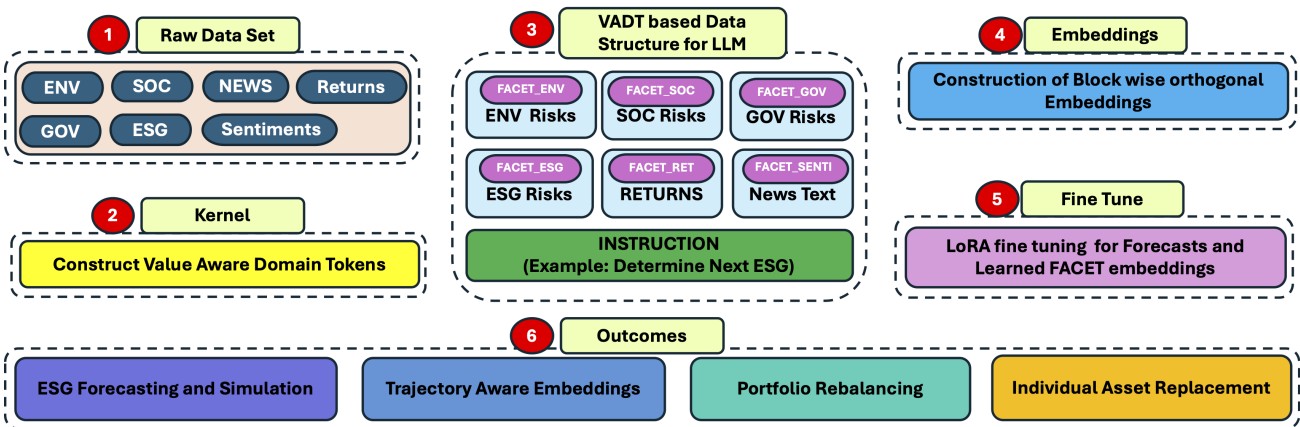

*Figure 1.* Block diagram of our ESG Domain Aware System.

tion. We evaluate our method on real, rich and diverse data covering S&P stocks and major exchanges including NYSE, NASDAQ, PNK, OTC, AMEX, and CBOE.

### 3.1. Motivation: Domain-Aware Token Geometry

LLMs form organized structures within their input embedding space and input tokens tend to cluster by type (Khatir et al., 2024). When LLMs are finetuned with relationally similar embeddings it outperforms most of the state of the art methods on most analogous questions. For instance, RelBERT is capable of modeling relationships that go beyond those just covered in training data (Ushio et al., 2025). Building on these findings we segregate tokens into different subspaces orthogonal to each other such that each subspace represents tokens of specific type for LLM to understand different time series trajectories per company. To effectively get trained on companies trajectory the model should understand that ESG Risk of say 61.21 is more similar to ESG Risk of 61.02 and less similar to ESG Risk of 41.32. Likewise, it is important for LLM to establish that if 61.21 represents the ESG Risk ($\downarrow$) and 61.02 represents financial returns ($\uparrow$) then both numbers are not similar even if they are numerically close. Establishing relationships among the input tokens and the trajectories will enable generative models to forecast time series as well as simulate ESG series given other sequences about the company. The standard pre-trained tokenizers like the one of GPT-2 split numerical values into multiple sub-tokens. Specifically, individual float values were split into multiple sub-tokens, thereby losing the numerical significance. An example is given below, the occurrence of symbol Ġ in the output denotes leading whitespace before a token:

*ESG Ratings for GEN:*

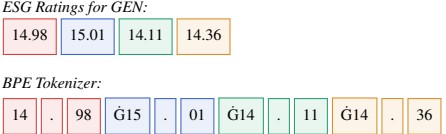

*BPE Tokenizer:*

| 14 | . | 98 | Ġ15 | . | 01 | Ġ14 | . | 11 | Ġ14 | . | 36 |

To effectively model temporal dynamics, the LLM must interpret numerical trajectories as structured sequences rather than plain text tokens, since treating numbers as ordinary text fails to capture their quantitative relationships and trends. Moreover, the model cannot interpret the type of numeric value (e.g., ESG risk or financial return) and its relative magnitude when compared to other values (e.g., relative to other peer companies). Both of them are critical requirements in ESG-finance applications. The problem is of even more importance when objective is to make LLM aware of trajectory of these domain tokens apart from just similarity between them. This presses a need of a value and domain aware LLM to produce accurate weights according to the geometry of the trajectory and maintain a correlation between output embeddings and input trajectories.

#### 3.1.1. VALUE AWARE DOMAIN TOKENS

A VADT is a token that encodes both its domain and value, and a sequence of VADTs represents the trajectory of the corresponding domain (eg., E/S/G). If two values $x$ and $y$ are of the same type $Q$ and are numerically close, then the embeddings of `<Q_x>` and `<Q_y>` should also be similar. We enforce this by shaping token embeddings so that numerically closer values of the same type receive similar representations. This requires both creating domain-specific tokens and preserving numerical relationships among them. We therefore encode all numerical variables as typed tokens of the form `<TYPE_VALUE>`. Examples include `<ESG_56.04>` for **ESG risk**, `<RET_12>` for **financial return**, with **ENV**, **SOC**, **GOV**, and **SENTI** denoting environment, society, governance, and sentiment types, respectively. We additionally use **ESGFO** and **ESGSO** to represent first- and second-order differences of ESG, enabling the identification of companies with similar velocity and acceleration toward the ESG objective.

**Embedding Construction:** Given a token of the form `<TYPE_value>`, this function maps it to a fixed-dimensional vector $\mathbf{v} \in \mathbb{R}^d$ using orthogonal, block-wise

sinusoidal encoding. The token prefix (e.g., ESG, RET, etc.) determines the block assignment. The embedding space is divided evenly into $n$ disjoint blocks as shown in Fig. 2.

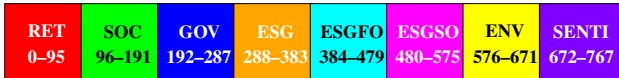

| RET | SOC | GOV | ESG | ESGFO | ESGSO | ENV | SENTI |
|-----|-----|-----|-----|-------|-------|-----|-------|
| 0–95 | 96–191 | 192–287 | 288–383 | 384–479 | 480–575 | 576–671 | 672–767 |

*Figure 2.* Partitioning of embedding vector into orthogonal spaces.

For a given prefix $p \in \mathcal{S}$, where $\mathcal{S}$ is the set of all token types given as, {RET, SOC, GOV, ESG, ESGFO, ESGSO, ENV, SENTI }, we employ

$$\text{Index of ESG in } \mathcal{S} = 3, \quad |\mathcal{S}| = 8, \quad b = \frac{d}{|\mathcal{S}|}.$$

This means the start index for ESG tokens is $3 \times b$. Suppose $d = 768$ (more details in supplementary), the block starting at start_idx $= 288$ is filled using sinusoidal encoding. This is inspired by transformer positional encoding (Vaswani et al., 2017). We modify the sinusoidal basis so that each block independently spans the full frequency range, ensuring consistent numerical representation across all tokens within that block. The major difference is that, (Vaswani et al., 2017) alternates sin/cos to build an orthogonal basis for positions which is good for sequences, but since ours is a token level embedding process we want monotonic similarity decay more than positional significance at this stage. The existing positional encoding and attention layers of LLM will still be used at a later stage anyways. This token level embedding style is added at input layer before positional encoding of LLM.

$$s_i(v) = \sin\left(\pi(x_i + \frac{v}{100})\right), \quad (1)$$

where $x_i$ is the normalized block size that assigns each dimension $v$ a unique phase, and $s_i(v)$ is the sinusoid-coded value of the corresponding numerical value. We also add a quadratic term to $s_i(v)$ that make embeddings differ as per numerical value. We compute this series for each token in the block and reposition the block in $\mathbb{R}^{768}$ at the right index. As a result, tokens of each type are embedded in its own orthogonal subspace, and the encoded value varies smoothly with the numeric content of the token. This technique ensures that the embeddings for each of the token types occupy distinct area within the embedding space and hence model can accurately differentiate between tokens of different types. We are augmenting the LLM with additional domain specific vocabulary (VADTs) and with an adaptive layer (LoRA) which is trained on trajectories of VADTs. We do not merge these weights with the LLM weights but rather have them supplementary to existing LLM weights. We use GPT-2 for demonstration but concept can be extended to any language model supporting low rank adaptation. (Keraghel et al., 2024) highlights GPT is usually a best choice to study when objective is to compare or improve multiple LLM models. It also highlights that there is no effective correlation between size of the model and the effectiveness of its embedding size. This further influences selection of GPT-2 for our study.

### 3.1.2. DOMAIN SPECIFIC EMBEDDINGS

This embedding is defined at the company level rather than at the individual token level. It corresponds to the learned hidden states of trajectory-specific FACET tokens for each sample (e.g., <FACET_ESG>, <FACET_ENV>, <FACET_SOC>, <FACET_GOV>). We obtain these representations by extracting the final-layer hidden states of the FACET tokens from the LLM after feeding the trajectories as input. To enable the LLM to capture the contextual semantics of the trajectories, we train the adaptive layers (via LoRA) using a combination of forecasting and geometry-preserving loss functions:

$$\mathcal{L} = \mathcal{L}_{\text{forecast}} + \beta \cdot \mathcal{L}_{\text{geometry}}, \quad (2)$$

where $\mathcal{L}_{\text{forecast}}$ is the standard next-token prediction loss of the LLM, and $\mathcal{L}_{\text{geometry}}$ encourages the model to match distances in the embedding space with Dynamic Time Warping (DTW) distances between the corresponding time series, with $\beta$ controlling their relative importance. We employ constrained DTW with a Sakoe–Chiba band for computational efficiency (Grzejszczak et al., 2022). This form of DTW further restricts excessive temporal warping and has been successfully applied to time series similarity tasks, including stock market trajectory analysis (Rakthanmanon et al., 2012; Tomaszewska et al., 2025). For each block, we measure pairwise cosine distances of FACET output hidden states and penalize their deviation from the corresponding DTW distances of the respective raw series. This encourages companies with similar temporal trajectories (e.g., Financial Return evolution) to cluster together within the corresponding semantic block of the final embedding space. We compute Spearman correlations per block to check how well the embedding distances in each block match the ground-truth DTW similarities. We extract a FACET embedding from the model's final layer for each company trajectory. We then align pairwise distances in this embedding space with DTW distances between the corresponding time series using a geometric loss. Our geometric loss also functions as a regularizer along with encouraging respective FACET embeddings of similar trajectories to be closed to each other in embedding space and dissimilar ones to be far apart. This jointly trains the model for next-step forecasting along with giving geometric understanding to the model. Let $\mathbf{E} \in \mathbb{R}^{N \times d}$ denote the pooled last-layer LLM embeddings for a batch of $N$ companies, where $d = 768$. We partition $\mathbf{E}$ into $B = 8$ semantic blocks corresponding to distinct financial domains in Fig. 2, and each block $\mathbf{E}_b \in \mathbb{R}^{N \times d_b}$ encodes a specific series type (e.g., ESG risks, returns, sentiment). Let $\mathbf{D}^{\text{DTW}} \in \mathbb{R}^{N \times N}$ be the

pairwise DTW distance matrix computed from the ground-truth trajectories. The block-wise geometry loss encourages the learned embedding distances to preserve the structure of the DTW distances:

$$\mathcal{L}_{\text{geometry}} = \frac{1}{B} \sum_{b=1}^{B} w_b \cdot \frac{1}{N^2} \sum_{i=1}^{N} \sum_{j=1}^{N} \left( D_{ij}^{(b)} - D_{ij}^{\text{DTW}} \right)^2, \quad (3)$$

where $B$ is the total number of embedding blocks, $w_b$ is the flag for block $b$ when set to zero will prevent that blocks geometric learning, $N$ is the number of trajectories in the batch, and $D^{(b)} \in \mathbb{R}^{N \times N}$ is the pairwise embedding distance matrix for block $b$. We also use Spearman correlation to measure whether the rank ordering of pairwise distances is preserved between the embedding space and the ground-truth DTW space. Our objective focuses on relative similarity structure i.e., whether companies that are close in the original space remain close in the embedding rather than on matching exact distance values.

### 3.2. ESG Forecasting

To forecast ESG/ENV/SOC/GOV at time $t$, the LoRA weights are derived by assembling training data set till time $t-1$ which covers mentioned trajectories as well as raw news. The model is tasked to predict one of the trajectories (based on instruction given to it by specifying target series) at time $t$. Further predictions are done using **Sliding window Logic** by shifting the prediction window each time during inferencing phase. A recent work (Taoussi et al., 2025) demonstrates the superiority of rolling forecasts. In our method, model is back tested to predict 20% values reserved for test data set with window size of w = 8 (the value is selected by finding autocorrelation on the dataset). The window is shifted each time by one step to generate the next prediction. Early stopping is implemented with 60% training data set, 20% validation and 20% test data set. Parameter efficient adaption on different trajectories for the companies enables LLM to understand the temporal across companies. To validate this, we consider a different task. (Bryzgalova et al., 2025) highlights missing financial data affects more than 70% of firms that represent about half of the total market cap. Our domain-aware LLM learns from multiple past trajectories and can forecasts a company's ESG trajectory also considering patterns in the other trajectories, it amply demonstrates that our technique understands the nuanced language of ESG.

**Incorporating Company News Sentiments:** Standard RoBERTa (Liu et al., 2019) trained on Twitter-style data is used to assess sentiment in real-time financial news sourced from LSEG Refinitiv, Yahoo Finance, Seeking Alpha and Market Watch. We identify ESG-relevant news by semantically matching articles to Global Reporting Initiative (GRI) topics using a Sentence-BERT model (`all-MiniLM-L6-v2`), full implementation details are provided in the supplementary material. It is to be noted that any other off the shelf method can also be employed for sentiment analysis and semantic match.

### 3.3. Superior Asset Population

Our implementation enables the generation of flexible queries to identify superior assets. One key advantage of a domain-aware system is that superiority criteria can be customized. For demonstration, we construct a superior asset population (SAP) of assets whose financial return trajectory is similar to a reference company and then forecast their sustainability to identify firms with lower predicted ESG risk. Similarly it can also be computed over individual E/S/G, or news sentiment trajectories, since our embeddings are learned in a block-wise manner. Let $x$ be a reference company with forecasted ESG risk as $\text{ESG}_x$. For companies with returns similar to $x$, the SAP of $x$ is:

$$\text{SAP}(x) = \left\{ c \in \mathcal{C} \,\middle|\, \text{ESG}_c < \text{ESG}_x \right\}. \quad (4)$$

Equation (4) demonstrates how the SAP mechanism can be used to recommend **individual asset replacements**. Our key contribution is the design of a sustainability-specific, domain-aware LLM that forecasts future sustainability trajectories and SAP consisting of assets with improved projected sustainability profiles. As a downstream application of SAP, we demonstrate its seamless integration with standard portfolio optimization frameworks. In practice, portfolio construction typically involves many constraints beyond sustainability alone. In this work, our objective is not to propose yet another portfolio optimization algorithm, but rather to show that the asset population generated by SAP can be directly consumed by existing optimization layers.

**Query using FACET embeddings:** This embedding space enables FACET-wise queries for identifying *performance-preserving peers*. For example, given a company $X$ in an investor's portfolio, we can retrieve companies that exhibit financial return trajectories similar to $X$ while simultaneously exhibiting ESG trajectories similar to another reference company $Z$ known to have superior sustainability characteristics. To support efficient similarity search over such trajectory-level representations at scale, we store per-FACET embeddings for each (company, series) pair in a vector database (our implementation uses Pinecone), enabling fast semantic retrieval for multiple downstream applications (portfolio optimization merely being one of them). To the best of our knowledge, this is the first LLM-integrated system that supports trajectory-level, performance-preserving peer discovery.

#### 3.3.1. DOWNSTREAM APPLICATION: CASE STUDY ON PORTFOLIO OPTIMIZATION

We formulate portfolio construction as a multi-objective optimization problem to demonstrate the integration of a

$\{(\text{ENV}, \text{ON}, \text{max}), (\text{SOC}, \text{ON}, \text{min}), (\text{GOV}, \text{OFF}, \text{min}),$
$(\text{ESG}, \text{ON}, \text{min}), (\text{Stock Diversity Risk}, \text{ON}, \text{min}),$
$(\text{Liquidity}, \text{OFF}, \text{constraint}), \ldots\}$

*Figure 3.* Example **configuration** used to assemble the portfolio optimization problem. The modular design allows sustainability objectives to be seamlessly added to financial objectives

sustainability-aware LLM with a downstream layer, rather than to propose a new optimizer. Let $w_i$ denote the allocation weight of asset $i$, with ESG risk $ESG_i$; our objective seeks to minimize the overall portfolio ESG risk under practical investment constraints using real data set,

$$f_1(x) = \sum_i w_i ESG_i + \lambda_1 \, \text{P}_{\text{return}} + \lambda_2 \, \text{P}_{\text{div}}, \quad (5)$$

where $\text{P}_{\text{return}}$ is a soft penalty that discourage lowering overall financial return compared to current portfolio and $\text{P}_{\text{div}}$ discourages allocation of stocks in small quantities. $w_i$ is the portfolio weight of stock $i$, $ESG_i$ is the ESG risk of asset $i$, and $\lambda_1, \lambda_2, > 0$ are trade-off parameters controlling the strength of these penalties (full equations for $\text{P}_{\text{return}}$ and $\text{P}_{\text{div}}$ are given in appendix). The objective jointly minimizes ESG risk while maximizing financial return, along with a diversity term preventing over-concentration in a single asset. Similarly, Eq. (5) can be specialized to construct objectives for the individual E, S, or G components. We validate this setup by applying standard multi-objective optimizers using SAP across different initial portfolios, observing stable convergence and consistently improved ESG profiles. To facilitate the integration of other constraints such as diversification, budget, liquidity, and transaction costs, etc., we adopt a modular, configuration-driven framework in which objectives are dynamically composed into the optimizers setup (Fig. 3). Objectives marked for maximization are converted to minimization by negation during setup, so the optimizer operates in a unified minimization form and computes a Pareto-optimal set of trade-off solutions.

## 4. Experiments

We first analyze different aspects of our method to highlight their strengths. Then we proceed to an extensive comparison with state of the art methods for ESG forecasting.

### 4.1. Numerical Closeness and Input Token Similarity

The cosine similarity between two tokens `<ESG_71.33>` and `<ESG_70.00>` is 0.99936, whereas between `<ESG_76.00>` and `<ESG_20.45>` it is 0.409. The similarity reduces as the numerical distance between the tokens increases. On the other hand, even when the numerical values are close, but the *type* is different, the similarity drops to zero. For example, the cosine similarity between `<ESG_71.33>` and `<RET_71>` is 0.0 (as they are orthogonal), which indicates a substantial difference in their embeddings. This is an advantage of having orthogonal

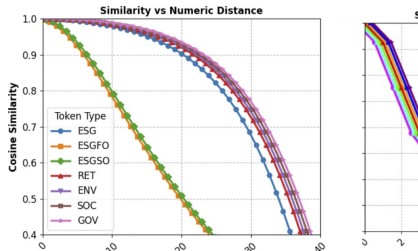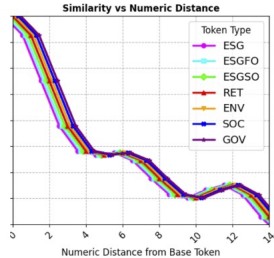

*Figure 4.* **Comparison of embedding strategies:** The left panel shows phased sine encodings, and the right panel shows alternate sine–cosine encodings. The phased sine approach produces a smoother variation in similarity over numerical distance, while the alternate sine–cosine method tends to flatten out, forming plateaus at certain distances

embedding spaces at input layer. Figure 4 left part shows cosine distances of our proposed token-level embedding and the right part shows for the widely used positional-level embeddings (Vaswani et al., 2017). Clearly our embedding shows the desired pattern preserving numerical significance, while latter is not following this relation (see plateaus in Fig. 4). Further, it shows variability between parameters as well, e.g., when velocity or acceleration drops the first order and second order tokens will show a sharp decline, which even fits better to define respective trajectory similarities.

### 4.2. Separation in Embeddings

This experiment demonstrates that tokens of different types are clearly distinguishable, as they occupy distinct regions in the embedding space. We trained a support vector classifier on 768-dimensional embeddings, with the resulting confusion matrix confirming that the model can uniquely classify each token type (see diagonal matrix in Fig. 6). When our domain-aware embeddings are removed, the ability to distinguish different tokens diminishes drastically (Fig. 6).

### 4.3. Input Trajectories & Domain Embeddings

Figure 5 shows that cosine distance between LLM embedding blocks corresponds closely to the DTW distance of the underlying trajectories. Using LLM we can now generate list of companies similar to specific company in one or more aspects (e.g., similar returns and similar ENV trajectory) and filter the set to identify superiority in other aspect (e.g., superior SOCIETY score). We also monitored the Spearman correlation on validation data set during adaptive layer training to verify that DTW based proximity between companies is preserved in the respective embedding block. With the designed loss function as a part of this work, the average Spearman coefficient across all blocks is 0.86, indicating strong alignment between DTW of trajectories and cosine distances of respective embedding blocks. For example, querying the return FACET embedding space for companies with trajectories similar to SYF using cosine similarity re-

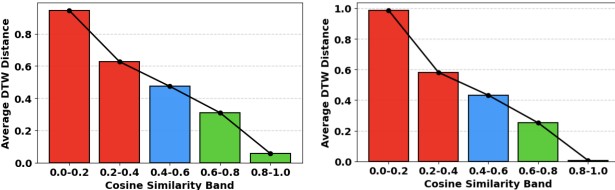

*Figure 5.* The figure illustrates that companies with more similar **ESG trajectories (left)**, with more similar **Return trajectories (right)** exhibit high cosine similarity between their respective FACET embeddings. The same is true for other trajectories like ENV, SOC, GOV. Our appendix gives more such examples.

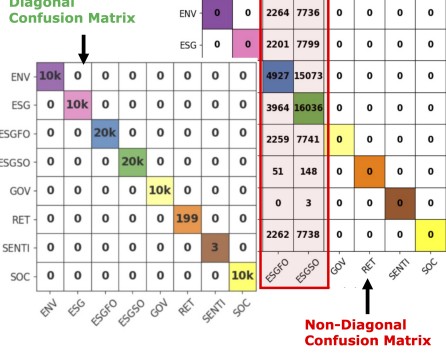

*Figure 6.* Confusion matrices comparing a support vector classifier with and without orthogonal subspaces. The diagonal matrix confirms clear token-type classification, while the non-diagonal matrix shows the loss of distinction without domain-aware embeddings.

trieves DHR (0.9896), FDS (0.9885), and NTRS (0.9885) as the top three matches, where the values denote cosine similarity; the relationship between cosine similarity and DTW distance is shown in Fig. 5. We also store these embeddings in a vector database to support efficient similarity search.

### 4.4. Role of LoRA and Orthogonal Embeddings

We exposed the LLM directly to VADT tokens without implementing Low Rank adaptation or training. This produced high MSE of $86.97$, reflecting the need for Low Rank Adaptation. We also used LLM directly on VADTs without creating orthogonal embedding space, and obtained a similar drop in performance. We LoRA-fine-tune the domain-aware LLM with instruction prompts↑ so that a single model can follow the instruction and forecast different sustainability risks. When LoRA adapter is trained **without geometric loss/FACET tokens** the ESG MSE increased to 8.866 with MAE of 2.221 and when trained **without news text** the ESG MSE increased to 27.643 with an MAE of 4.143. This proves the importance of our FACETs tokens, geometric loss and news text.

### 4.5. CASE STUDY: Optimization with SAP

As a downstream application of our framework, we present a case study on ESG-aware portfolio construction under financial constraints. We optimize Environmental, Social, and

Governance (E, S, G) objectives subject to budget and diversification constraints, with results reported in Table 2. We generate SAP using a domain-aware LLM and integrate it with standard multi-objective evolutionary optimizers, including NSGA-II, NSGA-III, SPEA-II, RNSGA-II, and AGE-MOEA (Martínez-Barbero et al., 2024). The purpose of SAP is to identify financially similar companies while enabling ESG improvement without sacrificing returns. Each method is evaluated on 25 portfolios. All SAP-based methods converge reliably, producing Pareto-optimal solutions, with a better ESG risk as shown in Table 2. The results are demonstrated for a seed portfolio with ESG risk 47.5 and a 1Y return of 14.8% . Among all methods, NSGA-II combined with Euclidean-distance selection achieves the lowest ESG risk while preserving financial performance. We further validate the approach on 30 synthesized portfolios, where NSGA-II consistently outperforms other methods. Figure 7 illustrates the solution sets and selections using both Euclidean and Nash criteria. This also enables the optimizer to account for future sustainability rather than relying solely on historical performance.

**Appending additional constraints:** To demonstrate the extensibility of our configuration framework, we introduce two additional constraints: (i) a transaction cost constraint (including spread, market impact, and commission) and (ii) a liquidity constraint (cost proportional to trading volume) and demonstrated the optimizers convergence using SAP, achieving lower ESG risk and higher returns (36.73 and 29.7, respectively, for the same example in Table 2).

**Without SAP:** We remove SAP and run NSGA-II directly, followed by ESG risk forecasting using the domain-aware LLM. This results in a substantially higher ESG risk of 46.5 for the new portfolio, indicating failure to achieve sustainability although satisfying financial constraints. These results highlight the critical role of SAP and the domain-aware LLM in enabling sustainable optimization under complex constraints.

| Metric | NSGA2 | NSGA3 | SPEA2 | RNSGA2 | AMOE |
|---|---|---|---|---|---|
| Init.Value | | | $28757.28 | | |
| ESG Risk ↓ | 37-39.5 | 39.2-40.2 | 39.25-40.75 | 38.7-39.0 | 38.5-40.0 |
| Returns↑ | 22.5-37.5 | 22-30 | 24-32 | 21.0-21.8 | 22-34 |
| Solutions ↑ | 471 | 16 | 471 | 471 | 471 |
| ESG-NASH ↓ | 39.67 | 40.17 | 40.59 | 39.01 | 40.23 |
| ESG-EUC ↓ | **37.1** | 39.1 | 39.6 | 38.66 | 38.6 |
| RET-NASH ↑ | 27.0 | 25.8 | 32.1 | 21.67 | 32.5 |
| RET-EUC ↑ | **37.46** | 23.23 | 31.03 | 21.81 | 23.55 |

*Table 2.* Convergence of multi-objective optimization models for ESG-based portfolio rebalancing for same initial seed portfolio.

### 4.6. Setting value of $\beta$

To encourage the LLM to capture the contextual structure of trajectories, we optimize a weighted sum of the forecasting loss and the proposed geometry-preserving loss (Eq. 2). We search $\beta \in [0, 1]$ and find that $\beta = 0.4$ yields the best test MSE on unseen data; this value is used for all reported

| Models | DA-LLM (Ours) | | iTransformer(2024) | | MM-iTransformer (2025) | | Chronos (2024) | | Sensorformer (2022) | | Reformer (2020) | | Autoformer (2025) | | TIMES-NET (2023) | | D-LINEAR (2023) | |
|---|---|---|---|---|---|---|---|---|---|---|---|---|---|---|---|---|---|---|
| Metric | MSE | MAE | MSE | MAE | MSE | MAE | MSE | MAE | MSE | MAE | MSE | MAE | MSE | MAE | MSE | MAE | MSE | MAE |
| Environment | **1.653** | **0.756** | 9.122 | 2.404 | 15.752 | 3.149 | 22.928 | *1.002* | 17.898 | 3.735 | 26.452 | 4.033 | 28.270 | 4.112 | 15.498 | 1.748 | 31.327 | 1.728 |
| Society | **5.582** | **1.912** | 13.668 | 2.971 | 22.483 | 3.891 | 22.984 | 3.505 | *5.777* | *1.960* | 7.553 | 2.216 | 20.241 | 2.697 | 12.061 | 1.989 | 23.546 | 2.106 |
| Governance | **4.514** | **1.867** | 8.838 | 2.436 | 39.891 | 5.898 | 26.101 | 2.392 | 7.866 | 2.168 | 7.064 | 2.171 | 22.376 | 3.743 | 15.007 | 1.985 | 25.287 | 2.310 |
| ESG | **5.227** | **1.912** | 7.334 | *2.211* | 18.363 | 3.598 | 18.140 | *2.154* | 7.844 | 2.300 | 18.053 | 3.377 | 10.966 | 2.367 | 21.612 | 2.949 | 26.213 | 2.733 |

*Table 1.* Performance comparison across time series forecasting baselines, covering **Deep Learning** , **Transformer based** and **LLM** based time series models. Our method **DA-LLM** achieves state-of-the-art performance on sustainability forecast.

results. Table 8 in appendix shows results obtained for ESG forecast hence we select most optimal $\beta = 0.4$. Same should be set to reproduce the results. The geometric loss also acts as a regularizer, too little weight leads to slight overfitting, while too much weight harms the forecasting objective.

### 4.7. Ablations

Table 3 summarizes the ablation experiments, where each component is removed individually to evaluate its contribution. As shown, every component plays a significant role in improving performance, and removing any component adversely affects the MSE.

| Setting | MSE↓ | MAE↓ |
|---|---|---|
| Full model (all components) | **5.227** | **1.912** |
| No FACET & No geometry loss | 8.866 | 2.221 |
| No news text | 27.643 | 4.143 |
| No sentiment | 38.56 | 4.87 |
| No GRI filtering | 17.289 | 3.672 |
| No LoRA (VADT only) | 86.97 | 6.97 |
| No VADT | 128.97 | 10.53 |
| Only FACET & No geometry loss | 9.321 | 2.765 |

*Table 3.* Ablation study showing the contribution of different components of the proposed framework. Lower values indicate better forecasting performance.

### 4.8. Forecasting Baselines and Results

We would like to note that ours is the first of its kind method of employing LLMs for ESG related applications. Since straightforward baselines are not available, we selected a broad set of baseline models to represent different schools of time-series forecasting. We compare against a broad and representative set of state-of-the-art time-series forecasting models including modern Transformer-based forecasters (iTransformer (Liu et al., 2024), MM-iTransformer (Mou et al., 2025), Autoformer (Wu et al., 2021), TimesNet (Wu et al., 2023), Sensorformer (Cheng et al., 2022), Reformer (Kitaev et al., 2020)), Chronos (Ansari et al., 2024) as large-scale foundation models for time series, and D-Linear as strong linear and decomposition-based model. In our problem, the model must jointly forecast over heterogeneous numeric signals (i.e., multiple fundamentally different financial quantities such as overall ESG risk, its Environment, Society, and Governance components, financial returns, and news text + sentiment) and semantically distinct signals (eg.,

numbers that represent different concepts and objectives) e.g., ESG = 60 versus Return = 60 have completely different economic meanings and should not be treated as similar. Existing multivariate forecasters treat all input channels as homogeneous real-valued sequences and therefore cannot represent these semantic distinctions. Moreover, our task requires preserving cross-company **trajectory geometry** as companies whose ESG, ENV, SOC, GOV, or return trajectories evolve in similar ways over time should be close in representation space which standard forecasting models do not enforce. In particular, tokenization-based foundation models such as Chronos discretize continuous values into bins via quantization, which destroys fine-grained metric structure and smooth neighborhood relationships. Our values are still mapped to continuous, smooth, distance-preserving embeddings using block-wise sinusoidal encodings, which preserve ordering and local similarity within each domain while keeping different domains (e.g., ENV vs. GOV) strictly separated in orthogonal subspaces. This structural mismatch is reflected in the results in Table 1, where our method consistently outperforms all baselines across ESG as well as individual ENV, SOC, and GOV risk forecasts.

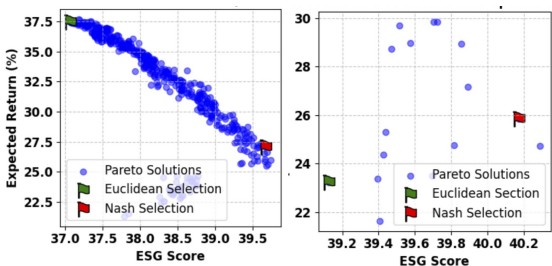

*Figure 7.* Portfolio choices offered by NSGA-II (left) and NSGA-III (right) using SAP.

## 5. Conclusion and Future work

The research proposed in this paper enables individual and institutional investors to contribute effectively toward responsible investment objectives. By combining value-aware domain tokens and trajectory-level FACET representations in a domain-aware LLM, our system achieves strong performance in forecasting ESG risks and identifying performance-preserving peers with lower sustainability risk, along with consistently improving downstream applications such as portfolio re-balancing. More broadly, this work shows how representation learning for temporal trajecto-

ries can be embedded to support sustainable investments. Looking ahead, this work provides a foundation for building sustainability-focused subject matter expert systems based on large language models. Such an expert assistant could interactively support investors in constructing performance-preserving sustainable portfolios through natural language queries. For example, an investor might ask: *"How can I improve my portfolio's ESG profile by replacing stock X while keeping stock Y unchanged?"*. Enabling this form of constraint-aware, conversational interaction represents a promising direction for making responsible investing more accessible and actionable in practice. Finally, exposing this system as an MCP server would enable a new class of agent and advisor-driven sustainability focused investments.Our long-term vision is to develop an intelligent agent that serves as a subject matter expert in ESG and sustainable investing, and can be integrated in existing financial ecosystem.

## Acknowledgements

The authors would like to acknowledge Saniya Yogesh Saratkar, Datta Meghe Institute of Higher Education and Research, for her valuable contributions toward the implementation and evaluation of some forecasting baselines used in this work. The authors also sincerely thank IdealRatings for their support during this research. We sincerely thank the Department of Artificial Intelligence, IIT Kharagpur for the generous support. The results and interpretations presented in this paper are that of the authors, and do not necessarily reflect the views or priorities of any organizations.

## Impact Statement

This paper presents a domain-aware AI framework for ESG forecasting and sustainability-aware financial decision support. The proposed system can assist investors in portfolio rebalancing, ESG risk forecasting, and identifying financially similar but more sustainable assets. By integrating ESG risks, financial returns, news, and sentiment into a unified representation learning framework, the model supports more informed and sustainability-focused investment decisions. The framework can further contribute toward improving accessibility to responsible investing for both individual and institutional investors. While AI-based financial models can significantly enhance analytical capabilities and support data-driven investment strategies, their outputs should be interpreted alongside human expertise, market knowledge, and broader economic considerations. Therefore, the proposed framework is designed to assist financial professionals and investors in making balanced and sustainability-oriented decisions, rather than functioning as a fully autonomous financial system.

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

# APPENDIX

The contents of this supplementary material are arranged as follows:

## A. Implementation Details

### A.1. Embedding Construction of VADTs

As mentioned in the main paper Fig. 2, we define the full embedding as a concatenation of eight block partitions. Each partition has a phase shifted sinosodial encoding within it. We consider 768D space as that is a default embedding space of GPT-2, while the analogy is extensible to other language models supporting low rank adaptation (Keraghel et al., 2024). We embed the 96D embedding for the token type at the respective block in 768D embedding space.

The combined block values are populated as,

$$b_i(v) = s_i(v) + q_i(v), \tag{6}$$

and block values are normalized (similar to (Vaswani et al., 2017)). $s_i(v)$ is the pivotal term, that is already discussed in main paper Eq (1). The quadratic term we used is:

$$q_i(v) = \frac{1}{2} x_i^2 \cdot \frac{v}{100}, \tag{7}$$

which grows with $x_i$ and the numeric value $v$. Here, $x_i$ represents the normalized index of the $i$-the position in the block. This ensures with each block we get smoothly changing values. The embedding space $\mathbf{E}$ can be viewed as list of blocks, where the embedding computed using Eq. (6) needs to now position itself into the main block. For the overall embedding $\mathbf{E}$ of 768D, each 96D block is embedded

in its respective slot as per Fig. 2. The embeddings are orthogonal to each other because the embeddings for a token of a specific type fall in a distinct block and gets positioned at specific index as shown in Fig. 2.

### A.2. Role of Sentiments in ESG Forecasting

To study the role of news sentiment, we conduct two experiments using the ESG generation task, where the ESG trajectory of a company is missing. In the first setting, we train the adaptive layer using *only* ESG trajectories from 60% of the companies, validate on 20%, and generate ESG trajectories for the remaining 20%. In the second setting, we train the adaptive layer using *both* ESG and news sentiment trajectories and again task it with predicting the ESG trajectories for the remaining 20% of the companies. We observe that incorporating sentiment signals leads to a substantial improvement in ESG trajectory prediction accuracy. This demonstrates the strong influence and practical relevance of news sentiment in modeling and forecasting ESG dynamics.

| Model | MSE $\downarrow$ | MAE $\downarrow$ | MAPE $\downarrow$ | Bias $\circ$ |
|---|---|---|---|---|
| W/O SENTIMENT | 38.56 | 4.87 | 8.45 | 3.89 |
| WITH SENTIMENT | 10.12 | 1.92 | 3.15 | 1.27 |

*Table 4.* Performance with and *without* Sentiment information. Clearly, sentiments are important for ESG forecasting.

### A.3. Exploratory Data Analysis

We have used real financial data sets from Yahoo Finance, FMP, and LSEG Refinitiv. Our data set assembled from these diverse upstream systems covers 6,21,408 real company samples with range of sectors including technology, industrials, energy, health care and more as shown in Fig. 10. Our data has real values for companies listed across S&P index and also companies listed on exchanges: NASDAQ, NYSE, CBOE, AMEX, PNK and OTC. To compute the news sentiments we have sourced real news, majorly from Yahoo finance, LSEG and also supplemented by Seeking Alpha and Market Watch channels through Finhub. We have calculated the sentiments over ESG relevant news articles and aggregated the sentiments by week to create a news sentiment time series. We have also sourced closed prices and computed last one year financial returns Month-on-Month to form return trajectory. Precision of our ESG risks and returns is upto 2 decimal places, this not only facilitates construction of VADTs and block wise embeddings with limited vocabulary tokens, but also avoids convergence based on baby steps of returns. Our dataset spans variety of real ESG, ENV, SOC and GOV values as shown in Fig. 11.

### A.4. Identifying ESG Relevant News

We filter and categorize news articles using the Global Reporting Initiative (GRI) sustainability reporting standards to retain only ESG-relevant news headlines for our model.

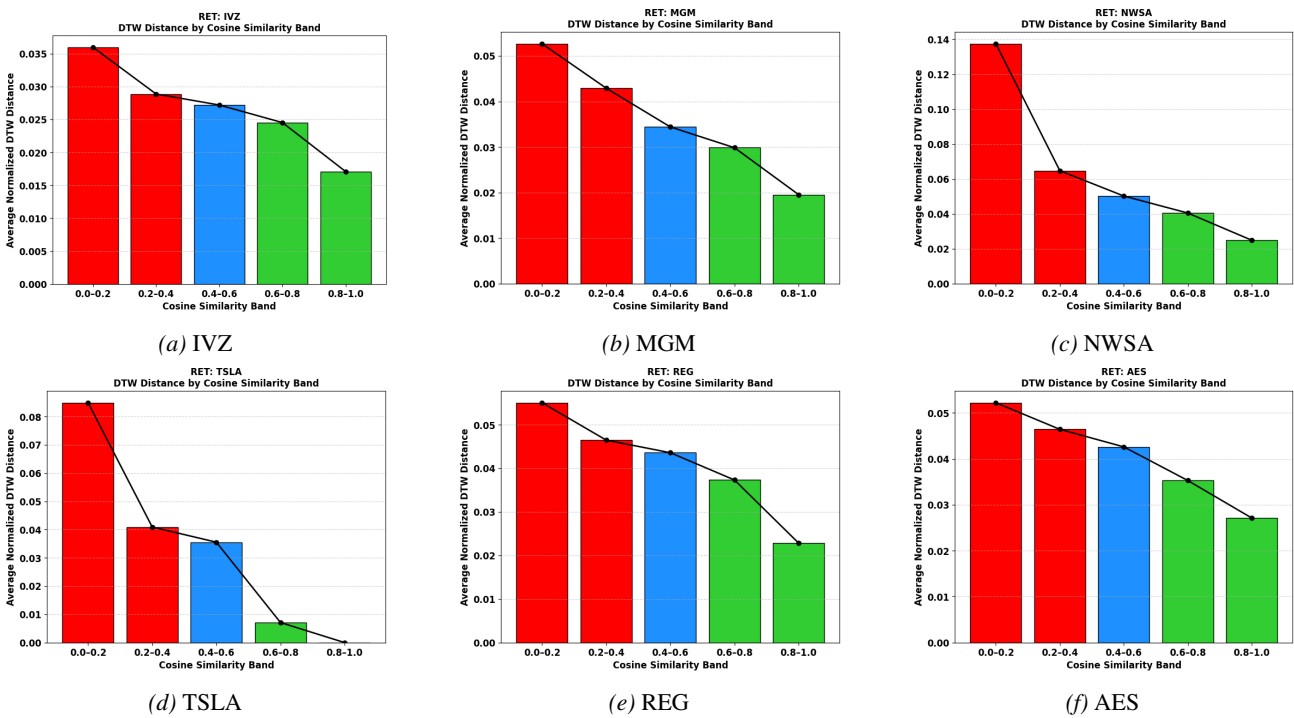

*Figure 8.* Relationship between **Financial Return Trajectories** and Similarity of Returns FACET Last-Layer Embeddings in Language Models. This figure demonstrates the examples of six companies. Clearly, when cosine similarity (x-axis) is high (or low) the DTW distance (y-axis) between the raw input trajectories is low (or high).

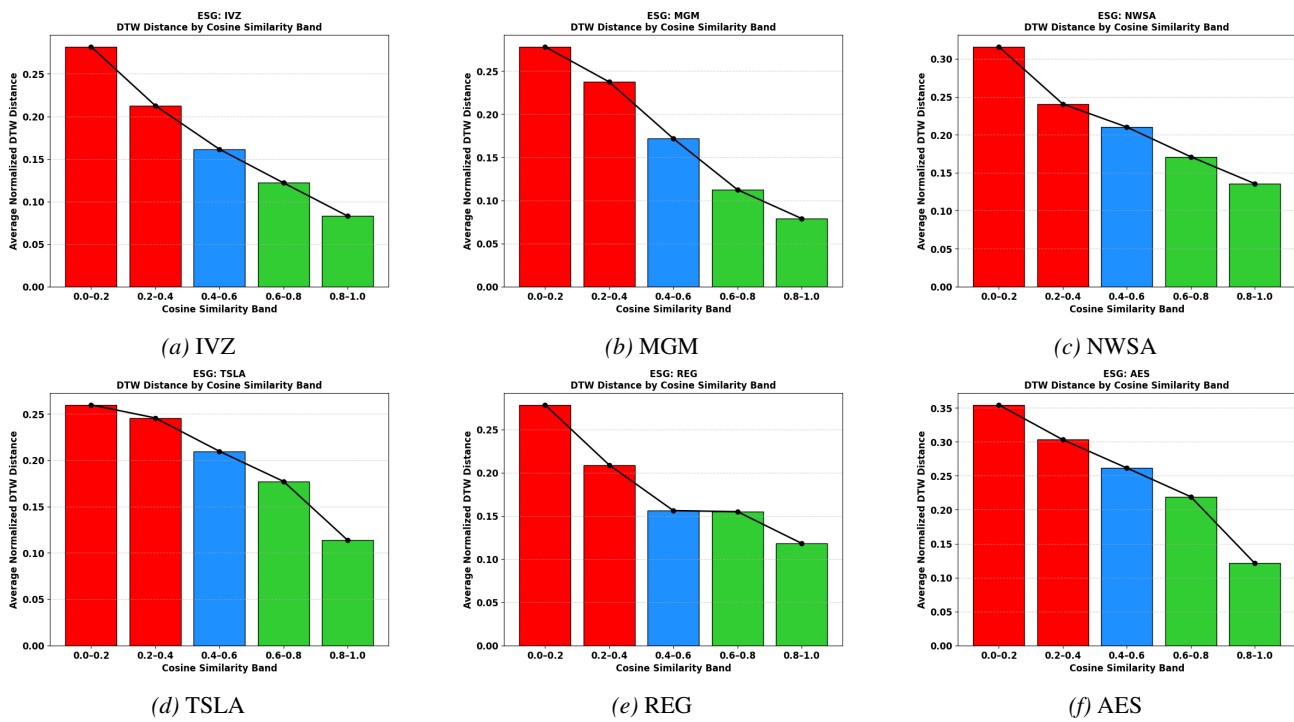

*Figure 9.* Relationship between **ESG Trajectories** and Similarity of ESG FACET Last-Layer Embeddings in Language Models. This figure demonstrates the examples of six companies. Clearly, when cosine similarity (x-axis) is high (or low) the DTW distance (y-axis) between the raw input trajectories is low (or high).

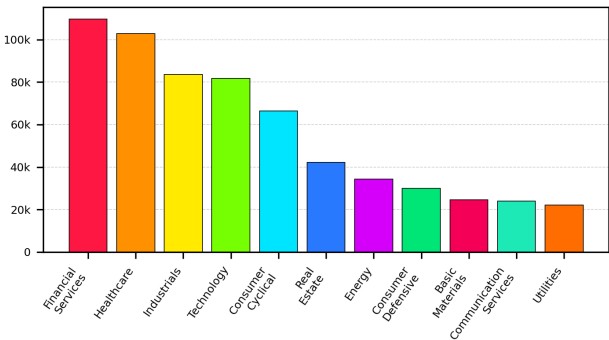

Figure 10. Range of Sectors in Data set. Our data spans diverse sectors.

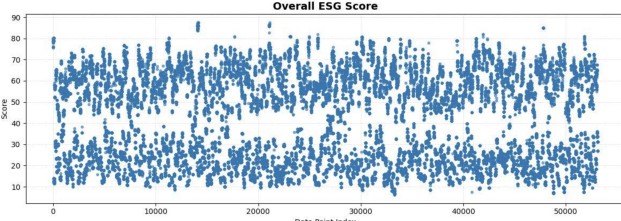

Figure 11. Range of ESG Risk scores in data set. Our data spans diverse ESG trajectories.

Each GRI topic (Table 5) is represented by its code and title, and each news item is represented by its headline or text. We use a pretrained Sentence-BERT. (Reimers & Gurevych, 2019) to assign each news article to most semantically similar GRI topic. This way we include only ESG relevant news in the context. Some examples include *"YUM! Brands Inc. - Sip, return, repeat: What one city's reusable cup trial taught Yum! Brands about sustainable packaging"* and *"Exxon Louisiana refinery poorly maintained, environmentalist"* maps to GRI 308 Supplier Environmental Assessment, *"Exxon unveils carbon-removal tech venture in green image push"* maps to GRI 305. We have mapping of each of the news articles identified in our data set with their GRI standard codes and this data is also available for further research. The news distribution is shown in 12. We also performed an ablation without implementing the GRI guided semantic matching and ESG MSE rose to 17.289 with MAE of 3.672, which proves the significance of guided semantic matching. We also identify the cosine similarity between news embeddings and filter news which are very similar to each other to avoid duplicate inclusion.

## A.5. Additional details on Case study: Optimization using SAP for Portfolio Re-balancing

As discussed in the main paper, superior asset population (SAP) enabled by our LLM model aid the portfolio re-balancing process. Unlike traditional methods that can re-balance portfolios using *only* the past data, our LLM allows to re-balance according to *both* past and future data, which we refer to as trajectories. This objective aligns to the need

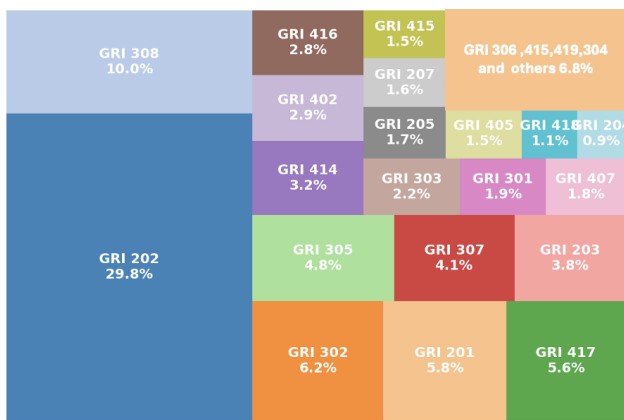

Figure 12. Distribution of real news text across GRI standards for Sustainability

| GRI code | Topic title | Pillar |
|---|---|---|
| GRI 201 | Economic Performance | G |
| GRI 202 | Market Presence | G |
| GRI 203 | Indirect Economic Impacts | G |
| GRI 204 | Procurement Practices | G |
| GRI 205 | Anti-Corruption | G |
| GRI 206 | Anti-competitive Behavior | G |
| GRI 207 | Tax | G |
| GRI 301 | Materials | E |
| GRI 302 | Energy | E |
| GRI 303 | Water and Effluents | E |
| GRI 304 | Biodiversity | E |
| GRI 305 | Emissions | E |
| GRI 306 | Waste | E |
| GRI 307 | Environmental Compliance | E |
| GRI 308 | Supplier Environmental Assessment | E |
| GRI 401 | Employment | S |
| GRI 402 | Labor/Management Relations | S |
| GRI 403 | Occupational Health and Safety | S |
| GRI 404 | Training and Education | S |
| GRI 405 | Diversity and Equal Opportunity | S |
| GRI 406 | Non-discrimination | S |
| GRI 407 | Freedom of Association and Collective Bargaining | S |
| GRI 408 | Child Labor | S |
| GRI 409 | Forced or Compulsory Labor | S |
| GRI 410 | Security Practices | S |
| GRI 411 | Rights of Indigenous Peoples | S |
| GRI 412 | Human Rights Assessment | S |
| GRI 413 | Local Communities | S |
| GRI 414 | Supplier Social Assessment | S |
| GRI 415 | Public Policy | S |
| GRI 416 | Customer Health and Safety | S |
| GRI 417 | Marketing and Labeling | S |
| GRI 418 | Customer Privacy | S |
| GRI 419 | Socioeconomic Compliance | S |

Table 5. GRI Topic Standards with codes.

for an adaptive portfolio rebalancing model to influence investment decisions (Iglesias-Casal et al., 2025). SAP gives the population for optimizer to select and converge to optimal portfolio. When re-balancing a portfolio for better-rated ESG, the superior asset population first gets an extract from LLM by getting companies with similar return trajectories as that of the current portfolio but superior sustainability forecast. We would reiterate that RETURN signal is used in our main paper in Fig. 5, while domain aware LLM can be generalized for any other trajectories, e.g., Fig. 9 shows the case for ESG trajectories.

**SAP Example:** In addition to what is discussed in main paper section 4.1.3, here we show another example of the output of SAP, which finds superior companies than EBAY. The latent space query engine first refers Return FACET embeddings to identify list of companies with similar return trajectory as that of EBAY and then identify those which

have lower ESG risk forecast (using DA-LLM). The format below is used to pass the SAP to optimizer. As seen below, 10 companies have similarity of 0.99-0.97 in their return trajectories but offer lower future ESG risk. This way, SAP can also be used for individual asset replacements as well, giving choices to investor based on similarity, if complete portfolio optimization is not an investment choice at this point in time.

```
EBAY ESG: 61.34   RET: 31.93 =>
(Alt.Company, Similarity, ESG, Return)
[('FFIV', 0.99, 48.08,35.93),
('WELL', 0.99, 59.95, 42.93),
('TMUS', 0.99, 53.70,  38.27),
('RSG', 0.99, 51.30, 33.0),
('BKNG', 0.988, 40.70,  39.2),
('GOOGL', 0.98, 60.71,  33.0),
('PNR', 0.98, 57.00,  36.8),
('GLW', 0.98, 58.78,  32.73),
('CTAS', 0.97, 53.63,  44.6),
('PANW', 0.97, 45.35,  40.6)]
```

Next we provide further details of the implementation of portfolio rebalancing based on SAP in Eq. (5). The term $\sum_i w_i ESG_i$, represents the portfolio-level ESG risk, computed as the weighted sum of individual asset ESG risks. The term $\lambda_1 \cdot P_{\text{return}}$ introduces a penalty for portfolios whose expected return falls below a predefined threshold. In our implementation we define the threshold ($\text{Ret}_{\text{thld}}$) as the return of initial seed portfolio. $P_{\text{return}}$ is given as

$$
P_{\text{return}} = \begin{cases} (\text{Ret}_{\text{thld}} - \text{Ret}(x))^2 & \text{if } \text{Ret}(x) < \text{Ret}_{\text{thld}}, \\ 0 & \text{otherwise.} \end{cases}
$$
(8)

The term, $\lambda_2 \cdot P_{\text{div}}$ ensures that all the funds are not allocated to single stock and sufficient diversity is maintained. The diversity constraint implemented for our experiments is,

$$
P_{\text{div}} = \sum_{i=1}^{n} \max\big(0, \text{Diversity\_factor} - \text{Shares}_i\big)^2, \quad (9)
$$

where $\text{Shares}_i$ denotes the number of shares allocated to asset $i$. Thus, $f_1(x)$ in Eq.(5) jointly minimizes the risk of portfolio ESG along with preserving returns, improving sentiment, and keeping fund allocation diverse. Similarly, $f_2(x)$, $f_3(x)$, $f_4(x)$ can be constructed as per focused E, S and G scores respectively. The design can be extended to incorporate custom objectives. While constraints like allocation diversity are not central to our problem, they reflect real-world optimization settings, and we show that the ESG objective still converges when optimized jointly with them. The weights can be chosen based on investor preferences. To reproduce the results in this paper, we set $(\lambda_1, \lambda_2) = (0.33, 0.40)$, selected after empirically testing

values from 0.1 to 1 and this setting gives max number of choices/solutions (471 for NSGA-II) for seed portfolios shown in Table 9(added at the end of appendix). We also confirmed that with all values our optimizer never suggests assets with higher ESG risk then the portfolio due to SAP.

## B. Analysis and Discussions

### B.1. Efficiency Statistics

We recorded efficiency statistics for our domain aware LLM on GPU A100, NVIDIA as shown in Table. All experiments were conducted on a single NVIDIA A100-SXM4-80GB GPU (compute capability 8.0, 85.17 GB total memory). The model contains 147,765,504 trainable parameters (0.591 GB), with an optimizer state size of 1.182 GB. The system achieves an estimated peak compute of 312 TFLOPs (FP16) and a peak throughput of 11,839.5 tokens/s (105.08 samples/s). During training, peak memory allocated is 5.86 GB, with maximum allocated and reserved memory of 9.82 GB and 11.65 GB respectively (25% memory utilization), while GPU utilization reaches 100%. The average power consumption during training is 298.1 W as shown in Table 6, **indicating that our implementation efficiently saturates the GPU compute while using only a small fraction of the available memory, leaving substantial headroom for scaling to larger models or batch sizes.**

| Category | Value |
|---|---|
| Model total parameters | 147,765,504 |
| Model trainable parameters | 147,765,504 |
| Model parameter size (GB) | 0.591 |
| Optimizer state size (GB) | 1.182 |
| GPU name | NVIDIA A100-SXM4-80GB |
| Compute capability | 8.0 |
| Visible GPUs | 1 |
| GPU total memory (GB) | 85.17 |
| Peak TFLOPs fp16 (est.) | 312.0 |
| Total peak TFLOPs (est.) | 312.0 |
| Peak throughput (tokens/s) | 11,839.5 |
| Peak throughput (samples/s) | 105.08 |
| Peak memory allocated (GB) | 5.86 |
| Peak memory reserved (GB) | 11.65 |
| Max allocated (GB) | 9.82 |
| Max reserved (GB) | 11.65 |
| GPU util. (%) | 100.0 |
| Memory util. (%) | 25.0 |
| Power (W) | 298.1 |

*Table 6.* Consolidated efficiency statistics.

### DTW Computation Timings

We evaluate the computational overhead introduced by the geometry-preserving DTW loss described in Section 3.1.2. Table 7 reports the average per-training-step runtime (forward + backward pass) for different batch sizes.

The additional DTW computation is modest and remains stable across increasing batch sizes, with $\Delta T$ varying only from 0.173s (batch size =32) to 0.186s (batch size =256). Moreover, the relative overhead decreases monotonically

as batch size increases, demonstrating that the constrained DTW component introduces a bounded and scalable computational cost.

| Batch | Without DTW (s) | With DTW (s) | $\Delta T$ (s) | Overhead (%) |
|---|---|---|---|---|
| 32 | 1.606 | 1.779 | 0.173 | 10.77 |
| 64 | 2.145 | 2.322 | 0.177 | 8.25 |
| 128 | 2.684 | 2.865 | 0.181 | 6.74 |
| 256 | 4.301 | 4.487 | 0.186 | 4.32 |

*Table 7.* Training overhead introduced by constrained DTW alignment.

### B.2. Changing timing of ESG forecast

The main paper Table 1 gives the latest results as of 2025, we also took the model back in March 2020 and attained MSE of ESG forecast sourced related company news from LSEG Refinitiv and other news sources. The MSE for ESG forecast is 5.68. We further did another test taking the model in Dec 2020, the MSE was infact slightly decreased to 5.22. We observed a slight increase in March followed by a stability in forecast in December. This prove our model is stable with time. We select the window size using the autocorrelation function (ACF), which yields a window size of 8. For robustness, we also evaluate window sizes 16 and 32, obtaining ESG MSEs of 5.816 and 5.912, respectively. Performance varies only marginally and remains superior to baselines, so we report results for the ACF-selected window size 8

### B.3. Search Space for Baselines

For each baseline model, we performed systematic grid searches. For each hyperparameter setting, we report per-target MSE as well as the average MSE across the four series. Here, we summarize for each model the exact hyperparameter ranges explored and the empirical performance ranges observed across the full grid. **Autoformer:** The search space includes `d_model` $\in \{64, 128, 512\}$, with `n_heads`= 8 and `e_layers`= 2. Over the grid, the observed average ESG MSE ranges from 10.966 (best) to $> 30$ (worst). **Reformer:** The search space includes `d_model` $\in \{64, 128\}$, with `n_heads`= 8 and `e_layers`= 2. Across all tested settings, the observed average MSE varies from 18.053 (best) to $> 26$ (worst). **MM-iTransformer:** The search space spans `d_model` $\in \{128, 256\}$, `n_heads` $\in \{2, 4\}$, fixed depth, `dim_feedforward` $\in \{512, 1024\}$, and `learning_rate` $\in \{10^{-3}, 5 \times 10^{-4}, 10^{-4}\}$. Across this grid, the observed ESG MSE varies widely, from 18.363 (best) to $> 60$ (worst). **iTransformer:** The search space includes $d_{model} \in \{64, 128, 256\}$, $n_{heads} \in \{1, 2, 4\}$, $dim_{ff} \in \{256, 512, 1024\}$, and learning rate $\in \{10^{-3}, 5 \times 10^{-4}, 10^{-4}\}$, with 3 layers, batch size 32, and dropout 0.1 fixed. Over this grid, the observed ESG MSE ranges from 7.334 (best) to $> 56$ (worst). **Sensorformer:** The search space includes $d_{model} \in \{64, 128\}$, $n_{heads} \in \{2, 4\}$,

$num_{layers} \in \{1, 2\}$, learning rate $\in \{10^{-4}, 5 \times 10^{-4}\}$, and dropout $\in \{0.1, 0.2\}$, with feedforward dimension fixed to 256 and additional Sensorformer-specific grouping and regularization parameters. Over this grid, the observed ESG MSE ranges from 7.844 (best) to $> 57$ (worst). **Chronos:** is fine-tuned via LoRA with r = 8, alpha = 32, dropout = 0.05. We use base model as it has equivalent 768D embeddings like ours. For quantization we used BIN_SIZE = 0.5 producing 201 bins. Experiments were also done with BIN SIZES 0.5,1.0 and 2.0. We have the best ESG MSE of 18.140 and worst $> 32.463$. The split per company is 60% training, 20% validation, and 20% testing. Similarly we have tried combinations for all baselines while we mention the key ones here.

### B.4. Details on additional Constraints added in Optimizer case study

As mentioned in the main paper section 4.5 we added 2 additional constraints using our constraint tool framework in optimizers algorithm and proved the convergence with better ESG risk. The objective of this paper is not to describe financial constraints but to prove that they can be added along with our SAP in the optimization layer. In this section we describe the implementation of additional constraints added to demonstrate this capability.

**Transaction cost penalty** Our transaction cost penalty calculates real trading costs in three parts: **Spread Cost:** 0.05% fee for buying/selling (the gap between buy and sell prices) **Market Impact:** Extra cost when trading large amounts that moves the price **Commission:** 0.1% broker fee with $1 minimum. Our implementation is as per (Li & Wang, 2013). Extending this concept we also implemented Liquidity penalty as below.

**Liquidity penalty**: Our liquidity penalty charges higher costs for hard-to-trade stocks. Stocks that trade over 10 million shares daily get only a 0.1% penalty because they are easy to buy and sell. Stocks trading 1-10 million shares get a 0.5% penalty. For 100,000-1 million shares, the penalty is 1.0%. Small stocks trading 10,000-100,000 shares get a 2.0% penalty, and very small stocks under 10,000 shares get the highest 5.0% penalty.

These are used in NSGA-II as penalty terms along ESG, ENV, SOC, GOV risk minimization and other objectives mentioned in main paper (5). The results in Section 4.5.1 show that, despite adding SAP-based penalties, the method still guarantees convergence while achieving the lowest possible ESG risk.

#### B.4.1. ADDITIONAL CONVERGENCE EXAMPLES

Optimization being a downstream application, the main paper gives only one example of convergence in Table 2. Here, we provide more examples demonstrating this capability in Table 9 (added at the end of appendix). Clearly, the results

show none of the algorithms elevated the ESG risk because of integration with SAP while NSGA-II with Euclidean distance outperformed with the given constraints offering lowest ESG risk and higher returns. Further, NSGA2 continued to offered more diverse choices and outperforms when paired up with Euclidean distance for portfolio selection, among the analyzed algorithms.

## C. Reproducibility

All results in this paper can be reproduced using following steps.

**Step 1: Environment Setup.**  Install the required dependencies using:

```
pip install -r requirements.txt
```

**Step 2: Data Preparation.**  Run the full data pipeline, which downloads and processes ESG data, stock prices, news text & sentiment (one XML file per company), peer information, and compiles the final dataset:

```
python run_all_pipeline.py
```

**Step 3: Model Training and Evaluation.**  Run the complete experiment pipeline:

```
python training_domain_aware_llm.py
```

This command trains the model and reproduces all forecasting, retrieval, and portfolio construction results reported in the paper.

**Step 3: Training and Testing.**  This step will generate all the results mentioned in the paper. It will start the model in evaluation mode and output predictions.

```
python eval_domain_aware_llm.py
```

The GIT repository for files is: **kunalp84/ESGGeometry**

### C.1. Setting Value of $\beta$ for Reproducibility

As discussed in Section 4.6 of the main paper, Table 8 reports the forecasting performance for different values of $\beta$. We observe that $\beta = 0.4$ provides the best trade-off between forecasting accuracy and geometric regularization.

| $\beta$ | 0 | 0.2 | **0.4** | 0.6 | 0.8 |
|---------|-------|-------|---------|--------|--------|
| **MSE** | 8.866 | 7.429 | **5.227** | 15.643 | 29.741 |
| **MAE** | 2.221 | 2.012 | **1.912** | 3.665 | 4.138 |

*Table 8.* Performance metrics for different values of $\beta$.

## D. Our Future Vision

In this work, we have demonstrated that domain- and trajectory-aware representations, combined with a domain-aware LLM, can outperform existing approaches in ESG forecasting and effectively drive portfolio rebalancing using sustainability-aware embeddings. Such a system would represent a new generation of sustainability-focused investment advices, capable not only of providing guidance on responsible investment choices but also of rebalancing portfolios and suggesting targeted asset replacements to improve both sustainability and performance with human in the loop for sufficient controls. This agent could operate either as an on-demand assistant or as a software-as-a-service (SaaS) advisory platform, significantly lowering the barrier to access for sustainability-aware financial decision-making and amplifying its potential social impact. By embedding sustainability considerations directly into everyday investment workflows, we believe such systems could influence and motivate regulators, individual investors, and institutional investors alike to prioritize sustainability as a core objective rather than a secondary constraint. In this sense, the work presented here lays the technical foundation for a broader shift toward making sustainable investing more accessible, actionable, and impactful in practice.

| Metric | NSGA2 | NSGA3 | SPEA2 | RNSGA2 | AMOE |
|---|---|---|---|---|---|
| **Init Portfolio 1** (Value=28757.28, ESG=47.5, Return=14.8) | | | | | |
| ESG Risk ↓ | 37–39.5 | 39.2–40.2 | 39.25–40.75 | 38.7–39.0 | 38.5–40.0 |
| Returns ↑ | 22.5–37.5 | 22–30 | 24–32 | 21.0–21.8 | 22–34 |
| Solutions ↑ | 471 | 16 | 471 | 471 | 471 |
| ESG-NASH ↓ | 39.67 | 40.17 | 40.59 | 39.01 | 40.23 |
| ESG-EUC ↓ | **37.1** | 39.1 | 39.6 | 38.66 | 38.6 |
| RET-NASH ↑ | 27.0 | 25.8 | 32.1 | 21.67 | 32.5 |
| RET-EUC ↑ | **37.46** | 23.23 | 31.03 | 21.81 | 23.55 |
| **Init Portfolio 2** (Value=8193.45, ESG=54.5, Return=24.8) | | | | | |
| ESG Risk ↓ | 43.0–47.0 | 45.0–47.0 | 44.0–47.0 | 46.98–47.03 | 44.0–48.0 |
| Returns ↑ | 32.5–50.0 | 36–42 | 34–46 | 34.06–34.18 | 34.0–46.0 |
| Solutions ↑ | 471 | 23 | 471 | 471 | 471 |
| ESG-NASH ↓ | 46.37 | 46.94 | 46.99 | 47.02 | 47.14 |
| ESG-EUC ↓ | **42.8** | 45.0 | 44.0 | 46.98 | 43.9 |
| RET-NASH ↑ | 41.6 | 42.0 | 41.2 | 34.13 | 42.8 |
| RET-EUC ↑ | **50.34** | 41.30 | 42.86 | 34.16 | 42.83 |
| **Init Portfolio 3** (Value=14961.00, ESG=48.5, Return=15.5) | | | | | |
| ESG Risk ↓ | 41.2–42.0 | 41.8–42.1 | 41.6–42.2 | 41.75–41.95 | 41.6–42.2 |
| Returns ↑ | 23–29 | 23.0–25.5 | 23.5–26.0 | 22.75–22.95 | 23.0–27.0 |
| Solutions ↑ | 471 | 21 | 471 | 471 | 471 |
| ESG-NASH ↓ | 41.98 | 42.04 | 42.16 | 41.96 | 42.19 |
| ESG-EUC ↓ | **41.2** | 41.8 | 41.6 | 41.74 | 41.6 |
| RET-NASH ↑ | 27.5 | 23.9 | 26.0 | 22.86 | 26.5 |
| RET-EUC ↑ | **28.58** | 25.85 | 24.67 | 22.97 | 25.72 |
| **Init Portfolio 4** (Value=6139.22, ESG=55.2, Return=30.6) | | | | | |
| ESG Risk ↓ | 43–47 | 45.5–47.0 | 44–47 | 46.35–46.55 | 44–47 |
| Returns ↑ | 32.5–50.0 | 34–41 | 34–44 | 32.65–32.85 | 32–46 |
| Solutions ↑ | 471 | 34 | 471 | 471 | 471 |
| ESG-NASH ↓ | 46.50 | 47.02 | 46.89 | 46.58 | 46.72 |
| ESG-EUC ↓ | **43.3** | 44.5 | 44.0 | 46.39 | 43.5 |
| RET-NASH ↑ | 40.7 | 37.9 | 40.6 | 32.69 | 41.3 |
| RET-EUC ↑ | **49.36** | 41.27 | 42.41 | 32.81 | 44.07 |
| **Init Portfolio 5** (Value=27074.60, ESG=54.1, Return=6.4) | | | | | |
| ESG Risk ↓ | 39–43 | 43.4–44.4 | 43.75–44.75 | 43.725–43.850 | 43.0–45 |
| Returns ↑ | 22–34 | 19.5–22.5 | 18–23 | 17.20–17.55 | 18–25 |
| Solutions ↑ | 471 | 20 | 471 | 471 | 471 |
| ESG-NASH ↓ | 43.15 | 44.51 | 44.78 | 43.83 | 44.91 |
| ESG-EUC ↓ | **38.3** | 43.6 | 43.9 | 43.72 | 43.2 |
| RET-NASH ↑ | 24.0 | 22.3 | 20.8 | 17.39 | 22.8 |
| RET-EUC ↑ | **34.69** | 20.68 | 20.51 | 17.52 | 21.67 |
| **Init Portfolio 6** (Value=13123.40, ESG=50.0, Return=11.5) | | | | | |
| ESG Risk ↓ | 39–42 | 41.0–41.8 | 40.25–41.75 | 41.66–41.74 | 40.0–41.5 |
| Returns ↑ | 20–70 | 20–32 | 20–32 | 19.15–19.35 | 20–45 |
| Solutions ↑ | 471 | 17 | 471 | 471 | 471 |
| ESG-NASH ↓ | 41.44 | 41.79 | 41.79 | 41.75 | 41.72 |
| ESG-EUC ↓ | **38.4** | 40.8 | 40.2 | 41.67 | 39.9 |
| RET-NASH ↑ | 29.9 | 25.0 | 22.8 | 19.18 | 22.8 |
| RET-EUC ↑ | **72.72** | 29.22 | 32.37 | 19.36 | 41.12 |

*Table 9.* Convergence of multi-objective optimization models for ESG-based portfolio re-balancing.

