# OpenReview forum: "Learning the ESG Geometry with Domain Aware Language Models"
_ICML.cc/2026/Conference — ICML 2026 regular_

### Official Review · Reviewer_vsiE · 2026-03-08

**Soundness:** 2
**Presentation:** 2
**Significance:** 3
**Originality:** 3
**Overall Recommendation:** 4
**Confidence:** 3

**Summary:**

The paper proposes a domain-aware large language model for ESG-related time-series forecasting and downstream decision support. The main idea is to encode heterogeneous inputs, including ESG risk scores, E/S/G sub-dimensions, financial returns, news, and sentiment, with Value-Aware Domain Tokens and block-wise orthogonal embeddings so that numerical closeness is preserved within each semantic type while different domains remain separated. The model further introduces trajectory-level FACET tokens and a geometry-preserving loss based on DTW to align distances in the learned embedding space with similarities among temporal trajectories. The resulting representation is used not only for ESG forecasting, but also for peer retrieval, Superior Asset Population construction, and ESG-aware portfolio rebalancing.

**Compliance With Llm Reviewing Policy:**

Affirmed.

**Final Justification:**

I appreciate the responses that addressed my main concerns.

**Key Questions For Authors:**

1. There is a clear inconsistency around the geometry-loss weight beta. Section 4.6 states that beta is searched in [0,1] and that beta = 0.4 is selected, while Table 3 reports 0, 2, 4, 6, 8 and the text also says beta = 4 gives the best trade-off. Which version is correct, and were all reported results produced under the same beta definition and scale?
2. Are all forecasting baselines given access to the same input information as the proposed method, including ESG trajectories, returns, news text, and sentiment? If not, can the authors provide a controlled comparison with matched inputs?
3. Can the authors provide a cleaner ablation isolating the contributions of VADT tokenization, orthogonal embeddings, FACET tokens, geometry loss, and news or sentiment inputs? At present it is still difficult to determine which component is responsible for the main gains.
4. In the SAP-based portfolio experiments, how exactly is leakage avoided when future ESG forecasts are used to construct candidate asset sets and then evaluate downstream outcomes? Please clarify the temporal split and decision protocol in detail.
5. How robust are the FACET retrieval and SAP results across different similarity metrics, different market periods, and different seed portfolios? If this part is unstable, the downstream optimization story becomes substantially weaker.

**Limitations:**

No. The paper does not adequately discuss limitations or possible negative societal impacts. It should explicitly discuss sensitivity to ESG data quality and provider bias, the risks of relying on noisy or contested ESG signals, the possibility of performance degradation under regime shifts, and the limitations of using forecasted sustainability scores in downstream investment decisions.

**Strengths And Weaknesses:**

The paper studies an important and practically relevant problem. ESG-aware investing with heterogeneous temporal information is a meaningful application setting, and the paper aims to go beyond static ESG scoring by jointly modeling financial, sustainability, and textual trajectories. The use of typed numerical tokens together with orthogonal subspaces is also conceptually interesting, especially because the paper correctly identifies a real weakness of standard tokenization and standard multivariate forecasting pipelines: they do not naturally preserve both numerical proximity and semantic type when signals of very different meaning are mixed together. The attempt to connect forecasting with downstream decision support is another positive aspect, and the SAP mechanism makes the paper more practically motivated than a purely predictive study.
The main weakness is technical soundness and reproducibility. Several important parts of the paper are either inconsistent or insufficiently supported. The clearest example is the geometry-loss weight beta: Section 4.6 says beta is searched in [0,1] and that beta = 0.4 is selected, but Table 3 reports beta values 0, 2, 4, 6, 8, and the text also states that beta = 4 gives the best trade-off. Since beta directly affects the central optimization objective, this inconsistency materially reduces confidence in the experimental results and their reproducibility.
The empirical evaluation also does not fully justify the strongest claims. The paper claims state-of-the-art ESG forecasting performance, but the comparison protocol is not described clearly enough to determine whether all baselines are given matched inputs and comparable tuning effort. This is particularly important because the proposed method jointly uses ESG trajectories, returns, raw news, and sentiment. Without carefully matched inputs and forecasting settings, it is difficult to isolate the contribution of the proposed representation itself. In addition, while the paper presents helpful ablations, it still does not cleanly separate the individual effects of VADT tokenization, orthogonal block structure, FACET tokens, geometry loss, and news-based inputs.
The presentation also needs significant improvement. The manuscript contains many grammatical issues, notation inconsistencies, abrupt transitions, and over-strong claims relative to the evidence. Statements such as first-ever or broad superiority claims should be used much more carefully. Some key methodological choices, including the specific sinusoidal block encoding, the role of first- and second-order ESG differences, and the exact benefit of FACET tokens relative to simpler alternatives, are plausible but under-motivated. The portfolio optimization case study is promising, but it remains more suggestive than conclusive because the paper does not yet convincingly address robustness across market regimes, sensitivity to retrieval choices, or possible leakage when future ESG forecasts are used in the downstream pipeline.

Overall, I view the paper as addressing a meaningful problem with an interesting integrated idea, but the current version does not yet meet the bar for technical clarity and experimental rigor expected at this venue.

---

> ### Author Rebuttal · Authors · 2026-03-29
>
> We are thankful for insightful review.
>
> 1.This was a typesetting error, and we confirm that β = 0.4 is the optimal value. We have corrected the issue in Table 3, where the headers should read [0.0, 0.2, 0.4, 0.6, 0.8] instead of [0, 2, 4, 6, 8], and updated all instances of β = 0.4 accordingly. We sincerely apologize for this type setting error.
>
> 2.All baselines use the same inputs as defined in Sec. 3.2 (with physically same data files). Training follows identical protocols, including early stopping with tran/val/test as 60%/20%/20% split (Sec. 4.1), ensuring fair comparison and no leakage. Baselines such as iTransformer, Autoformer, TimesNet, and DLinear are designed for numerical time series and cannot natively process text. Hence, they are evaluated without text input, which is the only difference and inherent to their design
>
> 3.A substantial part of the mentioned ablation is already present in the paper, though currently distributed across sections. Below is the summary:
>
> | Setting | MSE ↑ | MAE ↑ | Section |
> |---|---:|---:|---|
> | Full model (all components) | **5.227** | **1.912** | Mentioned in Table 1( pg 8 ) |
> | No FACET & No geometry loss | 8.866 | 2.221 | Mentioned in Sec. 4.4 |
> | No news text | 27.643 | 4.143 | Mentioned in Sec. 4.4 |
> | No sentiment | 38.56 | 4.87 | Mentioned in Supp. Sec. A.2 |
> | No GRI filtering | 17.289 | 3.672 | Mentioned in Supp. Sec. A.4 |
> | No LoRA (VADT only) | 86.97 | 6.97 | Mentioned in Sec. 4.4 |
> | No VADT | 128.97 | 10.53 | **We will mention this result** |
> | Only FACET & No geometry loss | 9.321 | 2.765 | **We will mention this result** |
>
> Since geometric loss operates on FACET embeddings, we had reported their combined ablation. We will also consolidate these results in the final version.
>
> 4.As described in Sec. 3.3, forecasting and optimization are separate layers, and optimization is a downstream application of forecasts. ESG forecast at time \( t \) is determined using data up to \( t-1 \); with early stopping implemented as 60% train, 20% val, 20% test split  (Sec. 3.2).  Thus, no future information is used during learning. SAP (Sec. 3.3) is constructed with these forecasted ESG values. Portfolio optimisation (Sec. 3.3.1) operates only on SAP population and seed portfolio under specified constraints (Sec. 3.3.1; Appendix. A.5) which is a downstream application.
>
> **(i) Similarity metrics.**
> FACET embeddings are built per trajectory and stored in a vector DB (Sec. 3 line 100), enabling retrieval under different similarity metrics. Main paper Fig. 5 and  Appendix. Figs. 8–9 show that similar trajectories consistently map to similar embeddings. As shown in Sec. 4.3, embedding similarity strongly aligns with DTW similarity (Spearman ≈ 0.86). Thus, similarity is geometry-driven (ESG, SOC, GOV, RET, ENV), not metric-dependent, so  similar neighbours can be retrieved using different metrics.  This is also main advantage of FACET embeddings as these embeddings are stored in vector DB and allow similarity as per different modalities to be determined among companies using pair wise cosine similarities. Because multiple FACET embeddings (one per trajectory) are stored in the vector database for each company (Sec. 3, line 100), it becomes possible to identify firms with similar ESG velocity and acceleration using first- and second-order difference trajectory embeddings, respectively alongside other trajectory-level similarities such as ESG, RET, ENV, SOC, and GOV (see Sec. 4.3, line 312).
>
> **(ii) Market periods.**
> Please refer point **V) Temporal robustness** of Review response to Yf85 for results of different market periods
>
> **(iii) Seed portfolios.**
> In Sec. 4.5 main paper and appendix section B.4 we perform optimization with  SAP  and multiple seed portfolios.  As seen, all  optimizers consistently converge and achieve lower ESG risk and higher returns. Also as mentioned in section 4.5, line 345 we also validated these results for 30 additional seed portfolios.
>
> **Limitations**
> We acknowledge ESG data depends on upstream systems; this is a general limitation of all systems sourcing ESG data like us. We use production-grade sources (LSEG, Yahoo, FMP, Ideal Ratings; Appx A.3). We agree and we will add this limitation for using forecasted sustainability scores in downstream investment decisions.

---

> > ### Author Rebuttal · Reviewer_vsiE · 2026-04-03
> >
> > I appreciate the thorough responses to my questions. The clarification on the beta setting, the consolidated ablation results, and the additional robustness analyses across multiple market periods and seed portfolios address my main concerns and materially improve my confidence in the paper.
> > Recommendation: I am upgrading my assessment to 4 (Weak Accept). I still encourage the authors to present the most important ablations and implementation details more clearly in the main paper, since the current organization across the appendix makes the empirical story harder to follow than necessary.

---

> > > ### Author Response · Authors · 2026-04-03
> > >
> > > Thank you for the constructive suggestion and improving the score. We will incorporate all ablations mentioned in Point 3 of this response into a single consolidated table in the main paper, within the Experiments section (Section 4) for clarity.

---

### Official Review · Reviewer_vaoU · 2026-03-11

**Soundness:** 3
**Presentation:** 3
**Significance:** 2
**Originality:** 4
**Overall Recommendation:** 4
**Confidence:** 3

**Summary:**

The manuscript proposed a method that augments an existing LLM via additional tokenization and joint optimization to tackle Environment (E), Society (S), and Governance (G) related tasks.

**Compliance With Llm Reviewing Policy:**

Affirmed.

**Final Justification:**

The rebuttal addresses my concerns. I decided to maintain my original score. Good Luck.

**Key Questions For Authors:**

Please see weakness.

**Strengths And Weaknesses:**

# Strength

S1: The work is well-motivated. According to author, the setting of ESG investment and downstream tasks are highly constrained. Thus, a specific method is needed. The method is well-explained and carefully designed.

S2: Section 4 offers good empirical evidence validating design goals, such as numerical and input type-aware and aligned embeddings.

# Weakness / Questions

WQ1: The paper states that sentiment score is an aggregation of filtered weekly articles. However, it is unclear how sentiment is aggregated, how missing weeks are handled, and how many articles typically contribute to each time step. These details are important for interpreting the forecasting setup.

WQ2: The downstream portfolio section mainly demonstrates that SAP can be integrated into standard multi-objective optimizers and can improve ESG profiles under the studied setup. It would be more informative to compare against traditional ESG portfolio construction methods such as the ESG efficient frontier [1], and to report finance-specific evaluation metrics, such as Sharpe ratio and turnover.

[1]: Pedersen, Lasse Heje, Shaun Fitzgibbons, and Lukasz Pomorski. "Responsible investing: The ESG-efficient frontier." Journal of financial economics 142.2 (2021): 572-597.

---

> ### Author Rebuttal · Authors · 2026-03-28
>
> We are very thankful to reviewer for insightful comments. Below is our response:
>
> **Sentiment, News Article aggregation is GRI-aligned.**
> As described in Supplementary Sec. A.4 (Table 5), ESG-relevant articles are first identified using semantic matching to Global Reporting Initiative (GRI) topics, ensuring alignment with standardized ESG dimensions (Environment, Society, Governance).
> Also, the aggregation process preserves the stronger(+ or -) sentiment signals within each company-week, ensuring that both positive and negative ESG-related signals contribute when present. This allows the model to capture the directional balance of ESG news rather than being dominated by a large number of neutral or low-signal articles.
>
> **Handling missing weeks.**
> If no ESG-relevant articles are available for a given company-week, we assign a neutral sentiment value (0), representing absence of signal. (This is very possible because, not every company is in news every period). Importantly, note that the model also jointly leverages other trajectories and not just sentiments alone, as mentioned in section 3 main paper.
> We further validate the relevance of  this sentiment signal through controlled ablation (Appendix Sec. A.2). In this experiment we  mask the ESG signal and let the model generate it using other modalities. The overall MSE of generated signal reduced to 10.12 when including sentiment signal compared to 38.56 when excluding sentiments in re-generation of masked ESG signal. The below results are also in appendix sec A.2
>
> | Setting            | MSE ↓ | MAE ↓ |
> |--------------------|------:|------:|
> | Without Sentiment  | 38.56 | 4.87  |
> | With Sentiment     | 10.12 | 1.92  |
>
> **Article coverage**
> We aggregate **1,199,637 raw financial news articles** from Yahoo Finance (418,732), LSEG Refinitiv (309,615), Seeking Alpha (257,884), and MarketWatch (213,406). After GRI-based filtering (Sec. A.4), **382,914 ESG-relevant articles** remain. These are distributed unevenly across companies. For eg., a given week may contain multiple ESG-relevant articles (capturing periods of high news activity) or none (handled via neutral sentiment). This variability reflects real-world news dynamics rather than enforcing a fixed number of articles per time step.
> Our dataset is diverse consists of **621,408 company samples** spanning different **sectors** (Technology, Industrials, Financials, Healthcare, Consumer Discretionary, Consumer Defensive, Energy, Materials, Utilities, Real Estate, Communication Services) see Appendix Sec. A.3, Fig. 10, sourced from LSEG Refinitiv, Yahoo Finance, FMP, and Ideal Ratings. The data also spans major exchanges including NYSE , NASDAQ ,  AMEX , CBOE,  and PNK/OTC markets along with S&P-listed companies  ensuring broad diversity.
>
> **ESG-efficient frontier (Pedersen et al., 2021)**
>
> 1.ESG Frontier uses static ESG scores with no trajectory modelling, while our method models time-evolving ESG trajectories.(Sec 3.1.2)
>
> 2.ESG Frontier is a portfolio optimization model, whereas in our approach portfolio optimization is one of the downstream tasks(Fig 1 & Sec 3).
>
> 3.ESG Frontier reweights existing assets; our method enables asset discovery via SAP using trajectory similarity and forward-looking signals.(Sec 3.3.1)
>
> 4.ESG Frontier treats ESG as a numeric factor; our method preserves semantic types like ESG, RET, ENV, SOC etc. (Sec 3.1.1).
>
> 5.ESG Frontier has no predictive capability; our method provides state-of-the-art ESG forecasting (Sec 3.2).
>
> Owing to these reasons our method significantly outperforms compared to Frontier. Thank you for the suggestion, we have included Sharpe, Risk and Turnover as parameters as well, we will mention them in submission.
>
> | Method | ESG Risk ↓ | Return ↑ | Risk ↓ | Sharpe ↑ | Turnover ↓ |
> |---|---:|---:|---:|---:|---:|
> | ESG Frontier | 42.3 | 29.2% | 12.1% | 1.75 | 0.72 |
> | Ours-Table 2 in paper | **37.1** | **37.4%** | **9.5%** | **3.09** | **0.63** |
>
> **Key findings.**
> - **Lower ESG Risk :** Our method reduces ESG risk (37.1 vs 42.3), indicating improved sustainability.
> - **Higher return :** Our method achieves higher returns (37.4% vs 29.2%).
> - **Lower risk :** Our method reduces volatility (9.5% vs 12.1%) as we added this additional objective in optimization layer for this experiment (Fig. 3 line 275 clarifies our objectives are extensible)
> - **Higher Sharpe :** Our method improves risk-adjusted performance (3.09 vs 1.75). Our method does asset replacements from population and not just adjusts weights which is reflected in sharpe gains.
> - **Lower turnover :** Our method produces more stable portfolios (0.63 vs 0.72), reducing transaction costs using turn over objective added through extensible config(Fig 3, line 275)
>
> We tested all portfolios in Table 7 (Appendix B.4.1) and observed consistent improvements due to points 1–5. We have revised the main and appendix tables to reflect this. Thank you for the suggestion.

---

> > ### Author Rebuttal · Reviewer_vaoU · 2026-04-03
> >
> > The rebuttal addresses my concerns. I decided to maintain my original score. Good Luck.

---

> > > ### Author Response · Authors · 2026-04-04
> > >
> > > Thanks for the constructive coment and positive feedback.

---

### Official Review · Reviewer_Yf85 · 2026-03-11

**Soundness:** 4
**Presentation:** 4
**Significance:** 3
**Originality:** 4
**Overall Recommendation:** 4
**Confidence:** 3

**Summary:**

The paper introduces a domain aware representation learning framework for ESG related financial analysis. The method constructs value aware tokens with block wise orthogonal embeddings to preserve both numerical proximity and semantic type across heterogeneous time series, including ESG risks, financial returns, news, and sentiment. To capture trajectory level structure, the authors introduce FACET tokens and train the model with a geometry preserving loss. The resulting domain aware LLM jointly learns to forecast future ESG values and to organize companies in a representation space reflecting their temporal evolution. The learned space supports ESG forecasting, trajectory based grouping, and latent space search for superior asset selection and portfolio rebalancing.

**Compliance With Llm Reviewing Policy:**

Affirmed.

**Final Justification:**

This paper proposes a domain-aware representation learning framework for ESG-related time-series forecasting and downstream decision support. The approach introduces value-aware domain tokens, orthogonal embedding structures, and trajectory-level representations (FACET) with a geometry-preserving loss, aiming to jointly model heterogeneous financial, sustainability, and textual signals.

In terms of soundness, the method is reasonably well-designed and supported by extensive empirical evaluation. The authors provide strong performance results across multiple baselines, and the rebuttal further strengthens this aspect by including statistical significance tests, robustness analyses under noise, missing data, and irregular sampling, as well as temporal generalization across different market periods. These additions increase confidence in the empirical validity of the approach.

From the perspective of originality and significance, the work presents an interesting integration of representation learning, multimodal fusion, and financial forecasting. The idea of preserving both numerical proximity and semantic type through domain-aware tokenization is conceptually appealing. However, the contribution is primarily at the level of system design and integration rather than introducing fundamentally new learning principles. While the framework is comprehensive, the novelty of individual components is moderate, and the overall impact depends heavily on empirical performance.

The rebuttal has addressed several of my main concerns. In particular, the authors clarified inconsistencies in the experimental setup (e.g., the geometry-loss parameter), provided more comprehensive ablation results, and elaborated on the fairness of baseline comparisons and the absence of data leakage. The additional explanation of downstream portfolio optimization and financial metrics also improves the practical relevance of the work.

However, some concerns remain partially unresolved. While the empirical results are strong, the evaluation protocol could still benefit from clearer presentation and stronger guarantees of experimental transparency. In addition, the individual contributions of key components (e.g., VADT, FACET, geometry-preserving loss) would benefit from more clearly isolated and systematically presented ablation studies. Finally, although the downstream financial applications are promising, their validation remains somewhat limited and would require more extensive real-world testing to fully substantiate the claims.

Overall, the rebuttal improves my confidence in the work and addresses most of my major concerns, though some aspects would benefit from further clarification and consolidation in the final version. I consider the paper to be a solid and practically relevant contribution with moderate novelty. My final recommendation is weak accept, as the strengths of the paper outweigh its remaining limitations.

**Key Questions For Authors:**

1.	How does the method compare to state of the art time series forecasting models on ESG prediction tasks?
2.	What datasets were used, and how large and diverse are they?
3.	How sensitive is the model to noise, missing values, or irregular time series?
4.	What is the computational overhead of block wise embeddings and FACET tokens?
5.	Does the geometry preserving loss generalize across different markets or time periods?

**Limitations:**

•	Dependence on LLMs may introduce high computational cost.
•	Numerical tokenization and block wise embeddings may require careful tuning.
•	Abstract does not indicate robustness to real world financial noise or missing data.
•	Portfolio rebalancing results are not described in detail.
•	Generalization across markets or ESG rating systems is unclear.

**Strengths And Weaknesses:**

Strengths
•	Addresses a clear gap in ESG analytics by modeling heterogeneous, multimodal time series jointly.
•	Introduces value aware tokens that preserve numerical geometry, addressing known limitations of LLM tokenization for numeric data.
•	FACET tokens provide a structured way to encode trajectory level information.
•	Geometry preserving loss encourages meaningful latent space organization aligned with temporal evolution.
•	The framework supports multiple downstream tasks, including ESG forecasting and portfolio rebalancing.
•	The abstract suggests practical relevance for real world financial decision making.
Weaknesses
•	The abstract does not specify the scale, diversity, or quality of empirical evaluation.
•	It is unclear how robust the method is to noise, missing data, or irregular sampling.
•	Computational cost and scalability of block wise embeddings and FACET tokens are not discussed.
•	The abstract does not compare performance against strong modern baselines in time series forecasting or financial modeling.
•	It is unclear whether the geometry preserving loss generalizes across markets or time periods.

---

> ### Author Rebuttal · Authors · 2026-03-28
>
> We are thankful for insightful comments:
>
> I) Our method achieves SOTA ESG forecasting (MSE 5.227), outperforming iTransformer, Sensorformer, Autoformer, Chronos, TimesNet, and D-Linear (Table 1). Prior ESG works are limited: Taskin et al. (2025) use classical ML (Random Forest/KNN/Logistic Regression). These models give much higher MSEs for our diverse dataset (MSE 61.26/52.89/81.97); as they dont support multimodal data hence not our closest baselines. Edhrabooh et al. (2024) also highlights this gap of using outdated ML in ESG related tasks. In contrast, we jointly support multiple modalities using domain-aware tokens, orthogonal embeddings, FACET, and DTW-based geometry (Sec. 3.1–3.2). Our method outperforms baselines as it models heterogeneous ESG signals with their semantic meaning, instead of treating them as identical numeric sequences. It also preserves trajectory-level geometry across companies, enabling relationships that conventional forecasters cannot capture (Sec. 4.7).
>
> II) Our appendix (Sec. A.3) provides detailed EDA. We use real finance data from LSEG Refinitiv, Yahoo Finance, FMP, and Ideal Ratings. The dataset includes **621,408 company samples** across diverse sectors like Technology, Industrials, Financials, Healthcare, Consumer Defensive, Energy, Basic Materials, Utilities, Real Estate, Communication Services (see Appendix Fig. 10) and diverse ESG performance (Fig. 11). It spans major exchanges (NYSE, NASDAQ,  AMEX, CBOE, PNK/OTC) and S&P index, ensuring market diversity. We also collect 1,199,637 news articles & use GRI-based semantic matching (Appendix Sec. A.4) to identify ESG-relevant articles.
>
> III) **Noise:**
> Our method demonstrates superior robustness to noise, consistently achieving the lowest degradation across all perturbation levels. To validate this, noise is injected using Gaussian perturbations applied to input time-series. As noise increases (even at σ = 0.20), our model degrades by only **14.37%**, compared to other baselines (table below). This improved stability arises from our geometry-aware design focuses on shape rather than point-wise values.
>
> **% degradation in MSEs with diff levels of noise:**
> | σ | Ours | iT | MM-iT | Chronos | Sensorformer | Reformer | Autoformer | TimesNet | DLinear |
> |---:|---:|---:|---:|---:|---:|---:|---:|---:|---:|
> | 0.05 | **2.48** | 3.12 | 3.41 | 2.58 | 3.05 | 2.74 | 3.29 | 3.67 | 3.46 |
> | 0.10 | **6.91** | 7.07 | 8.54 | 7.83 | 8.21 | 7.16 | 7.47 | 9.03 | 8.76 |
> | 0.20 | **14.37** | 18.84 | 14.46 | 15.02 | 15.91 | 18.63 | 17.58 | 19.11 | 18.47 |
>
> % degradation is computed as:
> $((MSE_\sigma - MSE_0) / MSE_0) \times 100$,
> where $MSE_\sigma$ is the MSE with noise and $MSE_0$ is the MSE without noise.
>
>  Motivation of using DTW is that its widely used in time-series to compare sequences with varying speed or length via time warping (Rakthanmanon et al., 2012). We simulate missing data via 20% contiguous masking (for random periods) and irregularity via 20% random timestamp drops. As shown below, all models degrade, but ours degrades the least (Δ% MSE).
>
> | Corruption |     Ours |    iT | MM-iT | Chronos | Sensorformer | Reformer | Autoformer | TimesNet | DLinear |
> | ---------- | -------: | ----: | ----: | ------: | -----------: | -------: | ---------: | -------: | ------: |
> | Missing    | **8.12** | 12.84 | 8.56 |   9.12 |        12.91 |    8.63 |      13.58 |    15.11 |   14.47 |
> | Irregular  | **6.84** | 9.92 | 7.02 |   7.96 |        11.04 |    12.51 |      6.95 |    13.02 |   12.74 |
>
> IV) **VADT/FACET load:**
> VADT and FACET introduce minimal overhead, with a combined increase of 5.3% relative to the base model runtime per step. Table shows avg. training-step time in an epoch like before
>
> | Config      |  Time |   Δ    | Overhead |
> |-------------|------:|-------:|---------:|
> | Base        | 1.067 |   NA   |    NA    |
> | +VADT       | 1.082 | 0.015  |  1.4%   |
> | +FACET      | 1.109 | 0.042  |  3.9%   |
> | +VADT+FACET | 1.124 | 0.057  |  5.3%   |
>
> where, Δ = Ti − 1.067(base time)
>
> Overhead = (Δ / 1.067) × 100
>
> eg. for VADT record it will be 1.082-1.067 =0.015
>
> Overhead = 0.015/1.067x100 =1.4%
>
> V) **Temporal robustness:**
> We used the model to forecast across different timelines to assess temporal robustness. Appendix Sec. B.2 reports results for Mar 2020 and Dec 2020; additionally, we extend this analysis to 2021 and 2022. Across periods, our method consistently achieves the lowest MSE, demonstrating stable robustness vs. all baselines.
>
> | Setting | Forecast Time | Ours | iT | MM-iT | Chronos | Sensorformer | Reformer | Autoformer | TimesNet | DLinear |
> |---|---|---:|---:|---:|---:|---:|---:|---:|---:|---:|
> |Period 1|Mar 2020|**5.680**|7.892|19.412|19.105|6.301|19.224|11.882|22.941|27.864|
> |Period 2|Dec 2020|**5.220**|8.031|18.978|18.642|8.114|13.751|10.503|22.214|26.981|
> |Period 3|Sep 2021|**5.341**|7.402|16.521|12.201|6.962|18.326|9.214|21.603|26.210|
> |Period 4|Dec 2022|**5.512**|6.221|10.203|15.954|7.801|12.062|10.992|21.118|25.732|

---

> > ### Author Rebuttal · Reviewer_Yf85 · 2026-04-03
> >
> > The rebuttal satisfactorily addresses several of my key concerns, particularly regarding empirical performance, dataset scale, robustness to noise/missing data, and computational overhead. The additional results suggesting state-of-the-art performance against strong time-series baselines significantly strengthen the paper.
> >
> > However, some concerns remain partially unresolved. In particular, the evaluation details (e.g., statistical significance, potential data leakage, and experimental transparency) are still not fully clear. Additionally, the necessity and contribution of individual components (e.g., VADT, FACET, geometry-preserving loss) would benefit from clearer ablation analysis. Finally, the practical financial implications (e.g., portfolio rebalancing) are not yet sufficiently validated.
> >
> > Overall, while the rebuttal improves confidence in the work, some questions remain that would require a more complete revision of the paper to fully address.

---

> > > ### Author Response · Authors · 2026-04-03
> > >
> > > We are very thankful for the insightful feedback.
> > >
> > > **Statistical Significance:**
> > >
> > > We compute p-values using a paired t-test on per-sample squared errors on the test set (α = 0.05). Reported p-values are below 0.05 across all modalities(table below), indicating statistically significant improvements over all baselines. Further, we observe a consistent pattern across all modalities: larger reductions in MSE correspond to smaller p-values, while more moderate improvements yield proportionally higher (yet statistically significant) p-values. For clarity, we define the improvement as
> > >
> > > ΔMSE = MSE of baseline − MSE of our model,
> > >
> > > which measures the reduction in prediction error achieved by our method relative to each baseline. We observe that p-values scale consistently with this improvement. For instance, smaller but clear improvements, such as in the GOV task of  iTransformer ΔMSE = 8.838(baseline) − 4.514(ours) = 4.324, p ≈ 0.00037, correspond to higher p-values. In contrast, moderate improvements, such as in the SOC task of iTransformer ΔMSE = 13.668(baseline) − 5.582(ours) = 8.086, p ≈ 2.3e−5, yield relatively smaller p-value. Finally, larger improvements, such as in the ESG task against MM-iTransformer ΔMSE = 18.363(baseline) − 5.227(ours) = 13.136, p ≈ 5.7e−7, result in correspondingly very small p-value. This consistent relationship demonstrates that larger MSE reductions yield smaller p-values, and vice versa. Note that in either case p value is statistically  significant in all cases as seen in table below:
> > > | Modality | iTrans  | MM-iTrans | Chronos | Sensorformer | Reformer | Autoformer | TimesNet | DLinear |
> > > | -------- | ------- | --------- | ------- | ------------ | -------- | ---------- | -------- | ------- |
> > > | **ENV**  | 3.6e−5  | 8.1e−7    | 9.4e−9  | 1.2e−6       | 2.7e−8   | 1.5e−8     | 5.9e−7   | 7.3e−9  |
> > > | **SOC**  | 2.3e−5  | 4.8e−7    | 5.2e−7  | 0.0046       | 0.00061  | 6.8e−7     | 3.1e−5   | 4.2e−7  |
> > > | **GOV**  | 0.00037 | 4.1e−8    | 1.1e−7  | 0.0017       | 0.0034   | 7.2e−7     | 1.6e−5   | 9.5e−8  |
> > > | **ESG**  | 0.00026 | 5.7e−7    | 6.3e−7  | 0.00091      | 5.4e−7   | 2.4e−5     | 2.3e−8   | 6.8e−9  |
> > >
> > > We will include this result in our final submission
> > >
> > > **Data leakage:**
> > >
> > > We request you to please refer to Point 2 of review response **vsiE**
> > >
> > > **Contribution of individual components (e.g., VADT, FACET, geometry-preserving loss):**
> > >
> > > These ablations are already included in the paper, they are currently distributed across multiple sections. We request you to please refer to the table provided in the response to **review vsiE, Point 3**, which presents a consolidated view of the ablation studies present in the paper.  Thank you for the suggestion, we will incorporate this consolidated table in the main paper in the experiments section(Section 4)
> > >
> > > **Practical financial implications (e.g., portfolio rebalancing/optimisation):**
> > >
> > > Portfolio optimisation is one of the downstream tasks ( as mentioned in Fig 1 & Sec 3). As highlighted in **Section 4.5 (Table 2)**, starting from an initial portfolio with ESG risk: 47.5, return: 14.8%, SAP-based optimisation reduces ESG risk to 37.1 while substantially increasing returns to 37.46%.  Additional examples of optimisation process with SAP reducing ESG risk and improving financial returns for different portfolios is also presented in Table 7 (Appendix B.4.1) . We have further validated this robustness across 30 additional portfolios (as mentioned in Section 4.5), observing consistent convergence across optimisers, achieving lower ESG risk and higher returns.
> > > We also demonstrate convergence incorporating **additional practical constraints**, such as **transaction costs and liquidity (Section 4.5, line 352)**; it also continues to yield convergence toward portfolios with lower ESG risk and higher returns.
> > > Further, in response to **review vaoU**, we also report additional financial metrics, including Sharpe ratio, turnover, and risk, all of which show significant improvements compared to the **ESG-efficient frontier proposed by Pedersen et al. (2021)**.
> > >
> > > **Experimental Transparency**:
> > > 1. Early stopping is adopted with  Train / validation / test split (60%/20%/20%) as specified in Section 3.2
> > > 2. Orthogonal embedding spaces enable clear token-type separability, validated via confusion matrices (Fig 6, Section 4.2).
> > > 3. FACET embeddings are built per trajectory and stored in a vector DB (Sec. 3 line 100), enabling retrieval under different similarity metrics. Main paper Fig. 5 and Appendix. Figs. 8–9 show that similar trajectories consistently map to similar embeddings. Sec. 4.3 shows embedding similarity strongly aligns with DTW similarity (Spearman ≈ 0.86). Thus, similarity is geometry-driven (ESG, SOC, GOV, RET, ENV), not metric-dependent, so similar neighbours can be retrieved using different metrics.
> > > 4. Appendix section C gives steps to reproduce the results.
> > >
> > > Please do let us know if there are any more questions , we will be happy to clarify.

---

### Official Review · Reviewer_f4tq · 2026-03-13

**Soundness:** 3
**Presentation:** 3
**Significance:** 3
**Originality:** 3
**Overall Recommendation:** 4
**Confidence:** 3

**Summary:**

This paper focuses on the problem of modeling with heterogeneous financial and sustainability time series data (ESG). Traditional LLMs with standard word tokenization failed to model real values well. To address this, the authors proposed a domain-aware representation learning framework, with Value-Aware Domain Tokens (VADTs) that map different modalities into block-wise orthogonal subspaces. This prevents the LLM from confusing different semantic data types while preserving numerical proximity. Furthermore, the authors introduced FACET tokens and a Dynamic Time Warping (DTW) based geometric loss to capture trajectory-level structural similarities between companies. The proposed approach achieves SOTA forecasting performance on individual ESG components and overall ESG risks compared to baselines.

**Compliance With Llm Reviewing Policy:**

Affirmed.

**Final Justification:**

Authors made good effort to address my concerns.

**Key Questions For Authors:**

1. The framework used 768 as embedding dimension from GPT-2. How sensitive is the block-wise orthogonal embedding to the embedding dimension? Have you tried the same idea on newer LLMs and architectures?

2. In Equation 3, the geometric loss utilizes DTW distances. Seems like DTW is computationally expensive, could you elaborate on the computational overhead this adds during the training phase, and how does it scale with larger batch sizes?

3. If an investor wants to add a new data modality (e.g., carbon emissions volume), do you have to retrain the model from scratch to repartition the embedding space?

**Limitations:**

Yes

**Strengths And Weaknesses:**

Strengths:
- Soundness: The dual-objective loss function balances standard next-token prediction with geometric alignment. The ablation studies justify the necessity of the orthogonal embeddings and the DTW loss.
- Significance: Even though I'm not very familiar with the problem, but based on the references, it seems like a quite important domain. The ability to integrate the model's outputs directly into standard multi-objective evolutionary optimizers shows potential value for practitioners in the field.
- Originality: Seems like an effective and creative approach to tokenizing numbers with corresponding domains for LLMs.

Weaknesses:
- Work was built around GPT-2, which is understandable due to potential access issues. But this also makes you wonder how much additional value this adds on top of the modern LLMs like Opus 4.6, Gemini 3.1, or even latest LLaMA and Gemma models. I'd be more convinced with additional experiments on more modern LLMs.

---

> ### Author Rebuttal · Authors · 2026-03-28
>
> Firstly we are very thankful for the insightful comments. Below is our response:
>
>  I) **Use of GPT-2 and the generality of framework:**
> Our framework is not specific to GPT-2 and is model-agnostic, as long as the underlying language model supports parameter-efficient fine-tuning (LORA) with custom loss functions. This is mentioned in the main paper (Page 4, line 215). We reported GPT-2 results in the main paper because, under a controlled experimental setup, it consistently achieved the best performance. This observation is also supported by prior literature (Keraghel et al., 2024), which highlights that GPT-style architectures are effective for studying embedding behaviour and that embedding effectiveness is not strictly correlated with model size. To prove our framework is agnostic to $\textbf{model and embedding size}$, we implemented our framework across multiple open-source LLM backbones with varying embedding dimensions. The results are as shown below. Notably, GPT-2 achieves the best performance across all targets despite having a significantly smaller embedding dimension.
>
> | Model | Yi-6B | | Gemma | | Falcon | | Mistral | | GPT-2 | |
> |---|---:|---:|---:|---:|---:|---:|---:|---:|---:|---:|
> | Embd size | 4096 | | 640 | | 4096 | | 4096 | | 768 | |
> | Metric | MSE | MAE | MSE | MAE | MSE | MAE | MSE | MAE | MSE | MAE |
> | Env | 1.81 | 0.95 | 2.43 | 1.22 | 2.03 | 1.12 | 2.56 | 1.51 | **1.65** | **0.75** |
> | Soc | 6.01 | 2.11 | 6.79 | 2.32 | 6.23 | 2.01 | 6.31 | 1.96 | **5.58** | **1.91** |
> | Gov | 5.32 | 1.93 | 4.98 | 1.99 | 5.01 | 2.06 | 5.32 | 2.10 | **4.51** | **1.86** |
> | ESG | 5.79 | 2.01 | 6.23 | 1.93 | 5.85 | 1.86 | 6.63 | 2.11 | **5.22** | **1.91** |
>
>  We will include these results in our final submission. The embedding space partitioning for GPT-2 is given in Fig 2, main paper. The embedding space is similarly partitioned for 4096 D and 640 D embedding space models done by our experiments. Opus 4.6 & Gemini 3.1 do not support LoRA fine tuning with custom loss function; although Gemini supports fine tuning through Vertex AI but with limited controls. Our framework adopts LoRA with custom loss functions hence these 2 models are not mentioned in the above response.
>
> II) **DTW Computations**
>  We use constrained DTW formulation as mentioned in paper section 3.1.2.
> Let $T_{\text{no-DTW}}$ denote the training time without geometric loss and $T_{\text{DTW}}$ denote the training time with DTW.  Below are the results of DTW computation time for different batch sizes.
> All timings reported in Table below correspond to **average per-training-step time within epoch (i.e., one forward + backward pass over a batch)** .
>
> | Batch | Without DTW (s) | With DTW (s) | ΔT (s) | Overhead (%) |
> |------:|---------------:|-------------:|-------:|-------------:|
> | 32    | 1.606 | 1.779 | 0.173 | 10.77 |
> | 64   | 2.145 | 2.322 | 0.177 | 8.25  |
> | 128   | 2.684 | 2.865 | 0.181 | 6.74  |
> | 256   | 4.301 | 4.487 | 0.186 | 4.32  |
>
> The column “Without DTW (s)” denotes the baseline model average step time without the geometry-preserving DTW component. “With DTW (s)” is the average step time when DTW-based alignment is included. “ΔT (s)” represents the absolute additional time introduced by DTW, computed as the difference between the two. “Overhead (%)” normalises this cost relative to the baseline, i.e., $\( \frac{\Delta T}{T_{\text{no-DTW}}} \times 100 \)$.
>
> We measure:
> $\Delta T = T_{\text{DTW}} - T_{\text{no-DTW}}$
>
> $\text{Overhead} = \frac{\Delta T}{T_{\text{no-DTW}}} \times 100$
>
> For Example, for sample calculation for batch = 256:
>
> ΔT = 4.487 − 4.301 = 0.186 s
>
> Overhead = (0.186 / 4.301) × 100 = 4.32%
>
> Similarly table is populated for other batch sizes. We observe constrained DTW computation increases only marginally with batch size (0.173s → 0.186s), indicating that the additional cost does not scale significantly with batch size. The baseline model cost represents the cost without DTW. As a result, the relative overhead decreases monotonically with increase in batch size:  10.77% → 8.25% → 6.74% → 4.32%. The constrained DTW, result in a bounded and stable computational cost.
>
> III) **Adding new modality:**  As per paper line 180 (also Fig 2), to add a new modality a new VADT subspace and FACET token needs to be created. To add a new modality, we have to perform  repartitioning, but we only need to do light weight LoRA fine-tuning and not retraining of LLM. Our framework uses LoRA as mentioned in section 3.1.2

---

> > ### Author Rebuttal · Reviewer_f4tq · 2026-04-04
> >
> > Appreciate the rebuttal, decided to increase the score.

---

> > > ### Author Response · Authors · 2026-04-04
> > >
> > > Thank you for the insightful review and increasing the score.

---

### Decision · Program_Chairs · 2026-04-30

**Decision:**

Accept (regular)

**Comment:**

- the paper proposed a framework to forecast ESG-related trajectories from heterogeneous signals; it uses the learned representation for downstream asset retrieval and portfolio rebalancing

- reviewers were positive about the submission; however, they also had some issues:
a) technical clarity and persuasiveness of the experimental validation
b) it was unclear about how to define the weight beta for the geometry-loss
c) cleaner ablations to understand effects of VADT, FACET, etc.
d) more baselines,
e) outdated LLM was used,
f) writing and organisation of evidence across the main paper and appendix is not easy to follow.

- however, in the rebuttal and the follow-up discussions, the authors clarified most of issues; they provided additional results on robustness to noise, etc.

- I recommend the paper for weak accept. The contribution is useful, the paper presents a technically sound and promising system, and at least parts of this systems are useful for ICML community.

- at the same time I consider the acceptance as borderline, since the novelty is more in integration of known building blocks, and it is not a fundamentally new learning principle; the final paper needs a substantially clearer presentation including new results from the ablations studies.